# The lysosomal transporter TAPL has a dual role as peptide translocator and phosphatidylserine floppase

Jun Gyou Park[1], Songwon Kim[1], Eunhong Jang[1], Seung Hun Choi[1], Hyunsu Han[1], Seulgi Ju[1], Ji Won Kim[2], Da Sol Min[1] & Mi Sun Jin[1] ✉

TAPL is a lysosomal ATP-binding cassette transporter that translocates a broad spectrum of polypeptides from the cytoplasm into the lysosomal lumen. Here we report that, in addition to its well-known role as a peptide translocator, TAPL exhibits an ATP-dependent phosphatidylserine floppase activity that is the possible cause of its high basal ATPase activity and of the lack of coupling between ATP hydrolysis and peptide efflux. We also present the cryo-EM structures of mouse TAPL complexed with (i) phospholipid, (ii) cholesteryl hemisuccinate (CHS) and 9-mer peptide, and (iii) ADP·BeF$_3$. The inward-facing structure reveals that F449 protrudes into the cylindrical transport pathway and divides it into a large hydrophilic central cavity and a sizable hydrophobic upper cavity. In the structure, the peptide binds to TAPL in horizontally-stretched fashion within the central cavity, while lipid molecules plug vertically into the upper cavity. Together, our results suggest that TAPL uses different mechanisms to function as a peptide translocase and a phosphatidylserine floppase.

TAPL (also referred to as the transporter associated with antigen processing-like [TAP-like] or ABCB9) belongs to the ATP-binding cassette (ABC) transporter superfamily[1]. It harnesses energy from ATP hydrolysis to translocate unwanted peptides from the cytoplasm into the lysosomal lumen for degradation[2]. TAPL is a homodimeric half-transporter composed of a transmembrane domain (TMD) and a cytoplasmic nucleotide-binding domain (NBD) with an N-terminal TMD (TMD0)[3,4]. TMD0 is made up of four transmembrane α-helices; it is dispensable for peptide transport but is essential for lysosomal targeting and long-term stabilization by interacting with the lysosome-associated membrane proteins, LAMP-1 and LAMP-2B[5,6]. TAPL has 38% and 40% sequence identity to TAP1 (ABCB2) and TAP2 (ABCB3), respectively[7,8]. The heterodimeric TAP1/TAP2 complex transports antigenic peptides from the cytosol into the endoplasmic reticulum, thus playing an essential role in loading peptides onto major histocompatibility complex (MHC) class I and triggering adaptive immune responses[9,10]. TAP complex favors peptides of 8–12 residues[11], whereas

TAPL transports a broad spectrum of peptides, ranging from 6 to 59 residues, with a preference for positively charged, aromatic, hydrophobic, or L-amino acids at both ends[2,12,13]. Despite its high sequence homology to the TAP complex, TAPL is not involved in the MHC class I-dependent pathway of antigen presentation[14], and high levels are found in testicular Sertoli cells, which function in lysosomal-mediated phagocytosis[7]. Furthermore, phylogenetic analysis indicates that TAPL homologs exist in organisms without adaptive immune systems, such as sea lamprey (*Petromyzon marinus*)[15,16] and nematodes (*Caenorhabditis elegans*)[17,18]. The two *C. elegans* TAPL homologs, HAF-4 and HAF-9, are essential for the formation of the lysosome-related organelles, gut granules[17,18]. Therefore, it can be predicted that TAPL is implicated in many fundamental cellular processes in lysosomes.

Recent studies have revealed the structures of human TAP1/2 in complex with the herpes simplex virus protein ICP47[19,20] and its bacterial homolog TmrAB from *Thermus thermophiles* in different functional states[21,22]. Nevertheless, details of the peptide binding

[1]School of Life Sciences, Gwangju Institute of Science and Technology (GIST), Gwangju 61005, Republic of Korea. [2]Department of Life Sciences, Pohang University of Science and Technology (POSTECH), Pohang 37673, Republic of Korea. ✉e-mail: misunjin@gist.ac.kr

mechanism of the TAP family are still unclear due to the lack of structural characterization of the complex with its natural peptide substrate. Here, we present the cryo-EM structures of mouse TAPL lacking the N-terminal TMD0 in (i) phospholipid-bound, (ii) both CHS- and 9-mer peptide-bound, and (iii) ADP·BeF$_3$-bound states at 3.2–4.0 Å resolutions. Combined with functional analysis, our structures provide a framework for visualizing the conformational changes of TAPL along the transport cycle and give insights into the mechanisms of peptide binding and its transport by TAPL. Moreover, to the best of our knowledge, this is the first report that TAPL exhibits lipid-dependent ATPase activity and transports fluorescently labeled phosphatidylserine from the outer to the inner leaflet of the proteoliposome in an ATP-dependent manner. Together, our results offer important clues for understanding the basis of the high basal ATPase activity of TAPL and the uncoupling between ATP hydrolysis and peptide efflux.

## Results

### Preparation of TAPL proteins for structural and functional studies

The mouse TAPL gene encodes a protein of 762 amino acids whose sequence is 94% identical to that of the human transporter (Supplementary Fig. 1)[7]. We generated recombinant baculoviruses expressing the full-length gene, and a truncated mutant lacking the N-terminal TMD0 (ΔTMD0). These proteins were overexpressed in *Trichoplusia ni* (Hi5) insect cells and purified with detergent micelles (DM, n-decyl-β-D-maltopyranoside) with and without CHS. We also reconstituted the proteins into nanodiscs composed of *E. coli* polar lipids. Gel filtration profiles revealed that ΔTMD0 eluted as a single monodisperse peak of the expected molecular weight (Supplementary Fig. 2). In contrast, the full-length protein exhibited significantly lower expression (data not shown) and eluted as a broad peak of large aggregates in gel filtration experiments. Previous studies have shown that the TMD0 of human TAPL folds independently from the core region[23] and can be deleted without any effect on ATP hydrolysis and peptide transport[5,13,24]. Given the above observations, we decided to concentrate our efforts on the ΔTMD0 construct to further investigate the structure–function relationship of TAPL.

### Functional characterization of mouse TAPL

The in vitro functional properties of purified ΔTMD0 were characterized using short synthetic peptides that are known to be transported by the human transporter[2,5,24]. ΔTMD0 preferentially transported 5-iodoacetamido fluorescein (5-IAF)-labeled peptides of 9 residues (RRYQNSTC$^{5-IAF}$L) in an ATP-dependent manner (Fig. 1a), and transport of longer peptide was significantly reduced, with 90% and 80% reductions for 18-mer (RRYQNSTELRRYQNSTC$^{5-IAF}$L) and 23-mer (RRYQNSTELRRYQNSTELNSTC$^{5-IAF}$L) peptides, respectively. ΔTMD0 also displayed as strong an uncoupling effect as human TAPL[24], that is, we did not observe peptide stimulation of ATPase activity at any concentration up to 250 µM (Fig. 1b). In controls, the presence of ATP analogs (ADP·BeF$_3$ or AMP-PNP), or a mutation conferring ATPase-deficiency (E664Q), prevented ATP hydrolysis in the presence of 9-mer peptide.

In NADH-coupled ATPase assays, the basal ATPase activity of DM-purified ΔTMD0 had a $K_m$ of $497 \pm 61$ µM for ATP, and $V_{max}$ of $394 \pm 10$ nmol/mg/min (Fig. 1c). The protein had an approximately 2- to 3-fold higher ATP turnover rate (951–1255 nmol/mg/min) in nanodiscs formed by POPS (16:0-18:1 PS, 1-palmitoyl-2-oleoyl-sn-glycero-3-phosphoserine), POPC (16:0-18:1 PC, 1-palmitoyl-2-oleoyl-sn-glycero-3-phosphocholine), or POPG (16:0-18:1 PG, 1-palmitoyl-2-oleoyl-sn-glycero-3-phosphoglycerol), whereas in nanodiscs made of *E. coli* polar lipids it's kinetic profile was similar to that in DM micelles (Fig. 1c). Curiously, although the same amount of protein was inserted into all the nanodiscs (Supplementary Fig. 3), no ΔTMD0 activity was detected in POPE nanodiscs (16:0–18:1 PE, 1-palmitoyl-2-oleoyl-sn-glycero-3-

phosphoethanolamine) (data not shown). We suppose that this is due to their structural instability because high concentrations of PE (>20 mol%) are notorious for causing incomplete bilayer formation[25]. Indeed, we observed that PE lipids formed large aggregates or liposome-like structures during nanodisc assembly. Taken together, these results demonstrate that mouse ΔTMD0 is a highly active homolog of full-length human TAPL, which means that it can serve as an excellent prototype for understanding how human TAPL works.

### Lipids as potential substrates for TAPL

Uncoupling of ATP hydrolysis from substrate transport has been reported in many ABC transporters[26,27]. Although its biological role and the underlying mechanism are still largely unknown, one suggestion is that these transporters catalyze the ATP-dependent transport of unidentified endogenous substrates, thus possessing high levels of ATPase activity in the absence of their known substrates[28]. Since ΔTMD0 exhibited unusually high basal activity in the reconstituted lipid systems (Fig. 1c), we speculated that the lipids might be substrates for TAPL.

To test this idea, we investigated the ability of various lipids to stimulate the ATPase activity of DM-purified ΔTMD0 and found that ΔTMD0 activity increased ~1.5–2.7 fold at saturating concentration (250 µM) of all tested lipids (Fig. 1d, e). ATPase activity was stimulated maximally by POPG, with $K_m$ and $V_{max}$ values of $105 \pm 16$ µM and $1147 \pm 41$ nmol/mg/min, respectively. In the case of CHS, which had the lowest stimulatory activity, the $K_m$ was unaffected ($127 \pm 28$ µM), whereas the $V_{max}$ was ~51% of that of the POPG-stimulated ATPase activity ($589 \pm 18$ nmol/mg/min). POPE-dependent ATP hydrolysis activity could not be measured because the addition of POPE to the ATPase reaction buffer caused nonspecific aggregation and rendered it opaque (data not shown). The stimulation of ΔTMD0 activity by lipid species supports our hypothesis that lipids are likely physiological substrates for TAPL.

### TAPL has ATP-dependent lipid floppase activity

Since many ABC transporters are known to have flippase (or floppase) activity[29,30], we examined whether ΔTMD0 in liposomes has such activity by monitoring the reduction of the fluorescence of 7-nitro-2-1,3-benzoxadiazol-4-yl (NBD*)-labeled lipids in the presence of dithionite ($S_2O_4^{2-}$) (Fig. 2a)[31,32]. Before proceeding with the flip–flop assays, we confirmed that the conjugation of NBD* chromophore to one of the acyl chains of phospholipids did not alter the dependence of ΔTMD0 ATPase activity on phospholipids (Supplementary Fig. 4). During liposome preparation using Bio-Beads, the residual detergent concentration was quantified by colorimetric analysis to determine the amount of Bio-Beads required to produce non-leaky proteoliposomes (Supplementary Fig. 5a, b)[33]. Two additions of Bio-Beads of 75 mg each, 150 mg total, completely removed detergent from both the buffer and liposome system. We also found that when we used liposomes composed of a 1:1 (mol/mol) mixture of *E. coli* lipids and egg PC containing 0.5% (mol/mol) NBD*-lipid, there was no subsequent decay of fluorescence despite multiple additions of excess dithionite, indicating that there was no leakage of dithionite across the liposome membrane (Supplementary Fig. 5c). In this article, we use an asterisk to differentiate the fluorescent NBD chromophore (NBD*) from the NBD domain of the protein.

All samples tested in the dithionite-based flip–flop assays exhibited similar two-step fluorescence decay curves (Fig. 2a). For example, in the presence of Mg$^{2+}$/ATP, the addition of 10 mM dithionite to ΔTMD-loaded proteoliposomes led to a fast drop in the fluorescence of NBD*-PS to ~36.2% of the original value (Fig. 2b). When a new equilibrium had been established, the addition of 1% Triton X-100 led to almost complete quenching of the NBD*-PS fluorescence. Under non-ATP hydrolytic conditions (Na$^+$/ATP), however, there was a more rapid initial reduction to ~31.7% of the original fluorescence upon exposure to dithionite (Fig. 2b). Thus, the difference between the inner leaflet fluorescence of proteoliposomes in the presence and absence

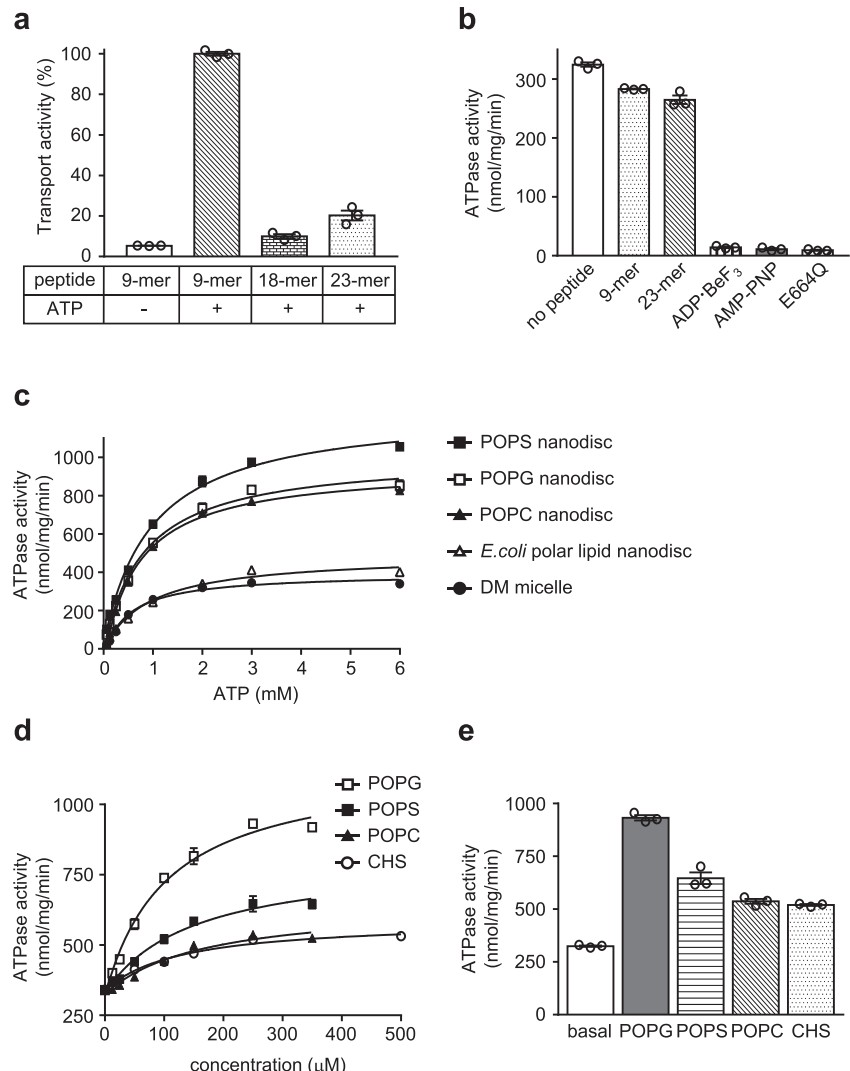

**Fig. 1 | Functional characterization of mouse TAPL. a** Peptide transport activity. Activity measured in the presence of 10 µM 5-IAF-labeled 9-mer peptide and 3 mM ATP was set at 100%. The data represent means ± standard error of the mean (SEM) of triplicate measurements of three different batches of proteoliposome preparations. Source data is provided as a source data file. **b** ATPase activities of DM-purified wild-type and the E664Q mutant in the presence of 250 µM peptides or ATP analogs (2 mM ADP·BeF₃ or 3 mM AMP-PNP). Each point is the mean of three replicate experiments using the same preparation. Error bars represent SEM. Source data is provided as a source data file. **c** ATPase activities of ΔTMD0 in DM micelles or various nanodiscs. Data are expressed as means ± SEM of three separate experiments using the same preparation. Source data is provided as a source data file. **d** ATPase activities of DM-purified ΔTMD0 as a function of lipid concentrations.

The $K_m$ of POPG was 105 ± 16 µM, and maximal ATPase activity was 1147 ± 41 nmol/mg/min (or 155 min⁻¹). The $K_m$ of POPS was 149 ± 36 µM, and maximal ATPase activity was 808 ± 44 nmol/mg/min (or 109 min⁻¹). The $K_m$ of POPC was 166 ± 45 µM, and maximal ATPase activity was 649 ± 35 nmol/mg/min (or 88 min⁻¹). The $K_m$ of CHS was 127 ± 28 µM, and maximal ATPase activity was 589 ± 18 nmol/mg/min (or 80 min⁻¹). Each data point is expressed as the means ± SEM of triplicate measurements from two different batches of protein preparations. Source data is provided as a source data file. **e** ATPase activities of DM-purified ΔTMD0 in the presence and absence of 250 µM lipid substrates. Each data point is expressed as the means ± SEM of triplicate measurements from two different batches of protein preparations. Source data is provided as source data file.

of hydrolysable ATP was ~4.5% (Fig. 2e). In contrast, when proteoliposomes were reconstituted with the catalytically inactive E664Q mutant, movement of NBD*-PS to the inner leaflet was reduced to ~1.9% (Fig. 2c, e). Accordingly, the true active floppase activity of the transporter was ~2.6% (Fig. 2e). It is noteworthy that translocation of ΔTMD0-mediated NBD*-PS was significantly inhibited in proteoliposomes containing 10% (mol/mol) unlabeled POPS (Fig. 2d, e). This result suggests that unlabeled POPS competes with its fluorescent derivative for transbilayer movement by ΔTMD0. Interestingly, translocation of NBD*-PS was also inhibited by the presence of the 9-mer peptide (Fig. 2b, e).

In our experimental conditions, ATP-dependent lipid transport by ΔTMD0 for NBD*-PS was high, whereas it was very low for PE, PG, and PC (data not shown), although the same amounts of protein were added to the liposomes (Supplementary Fig. 5d). Thin-layer chromatography (TLC) analysis also demonstrated that the failure to detect transport activity in the case of other phospholipids was not due to contamination with detergent-like lyso-phospholipids or other impurities (Supplementary Fig. 6). As mentioned above (Fig. 2d), this was probably because the PS species was the only one not already present in the liposome membrane (i.e. unlabeled lipids in the membrane compete with NBD*-labeled-lipids as substrates for ΔTMD0.). We also failed to detect ΔTMD0-mediated translocation of NBD*-cholesterol (data not shown). This may be because cholesterol undergoes rapid spontaneous movement (<1 min) across the liposome bilayer[34]. Consistent with the results of our dithionite-based flip–flop analysis, it has

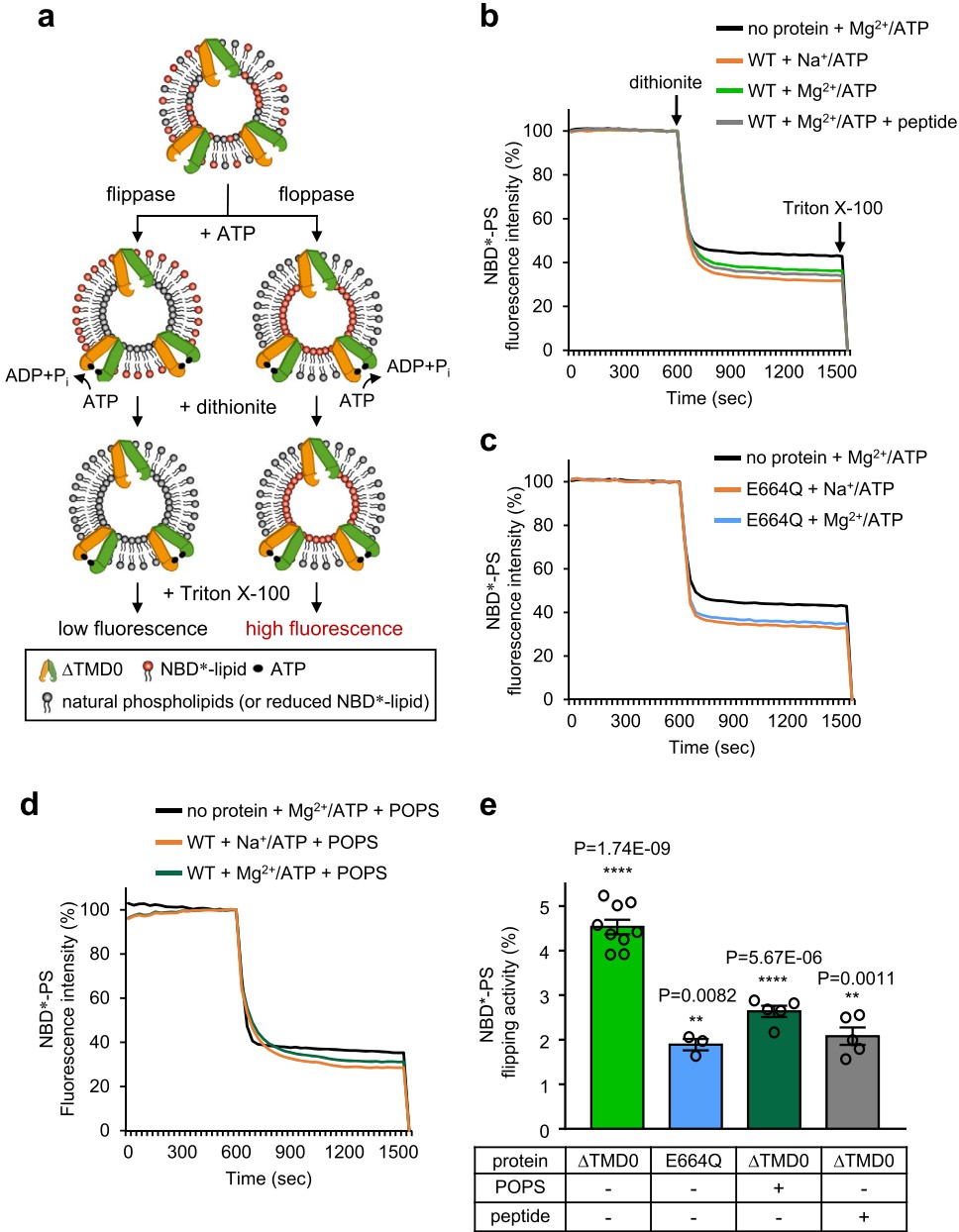

**Fig. 2 | ATP-dependent NBD\*-PS transport activity. a** Schematic of the fluorescence-based flip–flop assay used in this study. **b** Fluorescence traces of NBD\*-PS (18:1–06:0) floppase assays of WT-containing liposomes under various conditions. Results are the average of at least five measurements from two different batches of liposome preparations. Source data is provided as source data file. **c** Fluorescence traces of NBD\*-PS floppase assays of E664Q-containing liposomes. Each point is the mean of three replicate experiments using the same preparation. Source data is provided as source data file. **d** Fluorescence traces of NBD\*-PS floppase assays of WT-containing liposomes supplemented with 10% (mol/mol) POPS. Unlabeled PS inhibited ΔTMD0-mediated flipping of NBD\*-PS. Each point is

the mean of five replicate experiments using the same preparation. Source data is provided as source data file. **e** The net percentage of NBD\*-PS transported from the outer to the inner leaflet of proteoliposomes. The data represent the means ± standard error of the mean (SEM) of at least three independent measurements. The mark ** or **** represents the significant difference ($p < 0.01$ or $p < 0.0001$) between the percentages of NBD\*-lipid fluorescence measured in the presence of $Mg^{2+}$/ATP and $Na^+$/ATP in each assay. The *P* value was calculated by a two-sided unpaired *t*-test and adjusted using Welch's correction method. Source data is provided as a source data file.

been reported that insertion of hamster P-glycoprotein (ABCB1) into liposomes altered the distribution of NBD\*-lipids between leaflets of the membrane, with more NBD\*-lipids located in the outer leaflet[35]. This explains why we observed a greater decrease of NBD\* fluorescence in proteoliposomes than in protein-free liposomes (Fig. 2b–d).

## Overall architecture of ΔTMD0 and two separate substrate binding cavities

ΔTMD0 in nanodiscs was subjected to single-particle cryo-EM analysis in the absence of peptides and nucleotides. The protein particles were

seen to be evenly distributed in micrographs, and their front views displayed the typical structural fold of ABC transporters with two TMDs and two NBDs. We obtained a 3D reconstruction with no preferred orientation of the particles at an overall 3.4 Å resolution. Monomers of ΔTMD0 assemble to form homodimers in which each monomer contains a TMD with six membrane-spanning α-helices (TM 5–10) and a cytoplasmic NBD (Fig. 3a). The protein adopts the canonical inward-facing conformation in which the transport pathway opens to the cytosol and the two NBDs remain separate (Fig. 3b and Supplementary Fig. 7a, b).

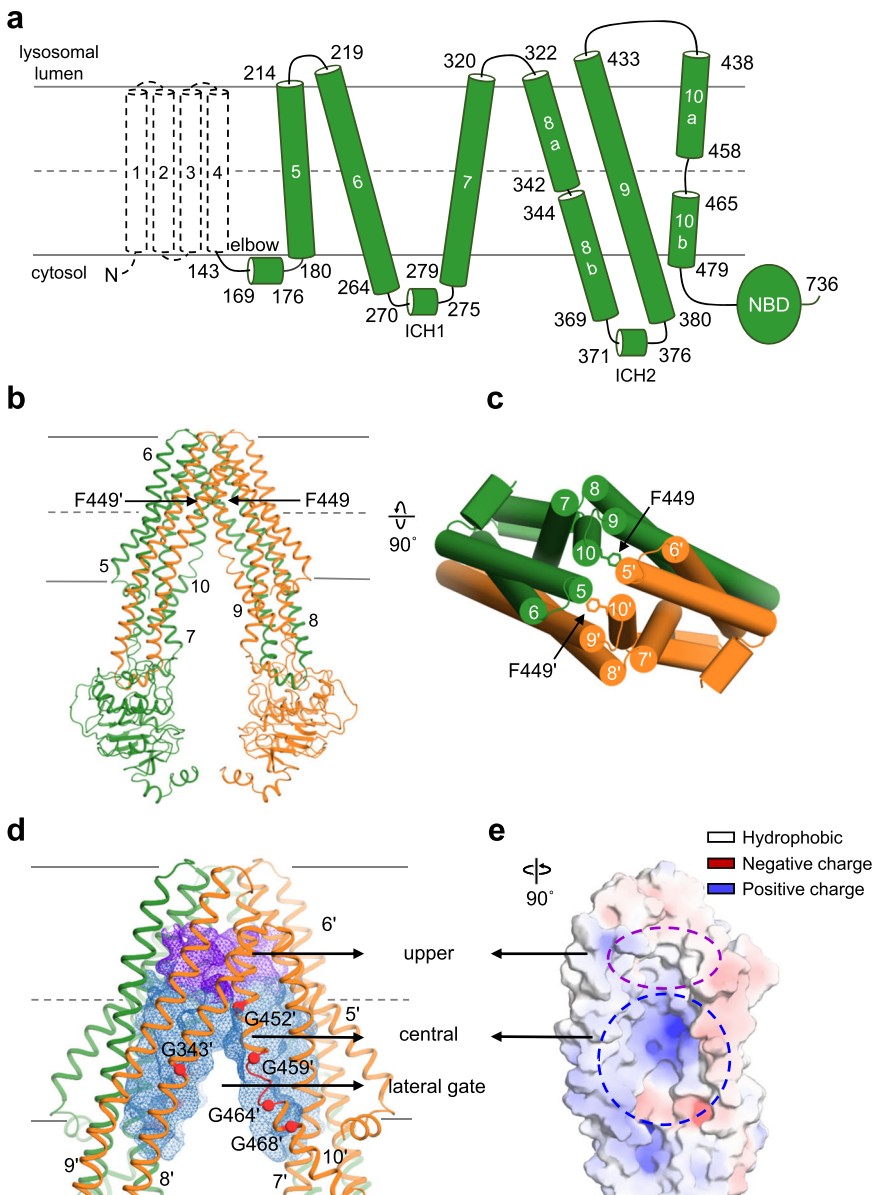

**Fig. 3 | The overall structure of nanodisc-reconstituted ΔTMD0 in the inward-facing conformation. a** Schematic representation of the mouse TAPL monomer. Dashed lines represent amino acids not included in the final construct. **b** and **c** Cartoon or cylinder representation of inward-facing ΔTMD0 in **b** front and **c** top view. The two monomers are shown in green and orange, respectively. The F449 residue in each monomer is shown as a stick. Single primes are used for the residues (or helices) of one monomer to differentiate them from those of the other monomer. **d** The substrate translocation pathway of ΔTMD0. The upper and central cavities are represented by purple and blue mesh, respectively. The glycine residues flanking the lateral opening are shown as spheres. The loop in TM 10 is colored red. **e** The substrate transport pathway is split along the 2-fold axis and rotated by 90°. The electrostatic surface potential of the substrate transport pathway was calculated using the APBS/PDB2PQR software suite, assuming a pH of 7.5 (http://www.poissonboltzmann.org). The surface electrostatic potential is contoured from −10 kT/e (negative, red) to +10 kT/e (positive, blue). The upper and central cavity regions are indicated by purple and blue dashed lines, respectively.

The structures of many inward-facing ABC exporters have an inverted V-shaped substrate translocation pathway with a large cytoplasmic opening and gradual narrowing toward the apex[36–38]. Accordingly, a major binding site for the substrate is inside the membrane inner leaflet when the transporter adopts an inward-facing conformation. However, ΔTMD0 has a cylinder-shaped transport pathway in which the highly conserved F449 residues on TM 10 penetrate into the transport pathway and face each other, forming the narrowest point (7.5 Å) (Fig. 3c). Consequently, ΔTMD0 has two separate substrate binding sites, a central cavity in the cytosolic leaflet of the lysosomal membrane and an upper cavity in the luminal leaflet (Fig. 3d). The central cavity is relatively hydrophilic because of the presence of many polar residues (Fig. 3e). By contrast, the upper cavity

is completely lined with hydrophobic residues, and is large enough to accommodate lipid molecules (Supplementary Fig. 7c). Similarly, human ABCG2 (a multidrug transporter) and ABCB6 (a porphyrin transporter) also contain central and upper cavities separated by a leucine (L554) plug, and a long bulge loop on TM 7, respectively[38,39]. However, due to the small volumes of the upper cavities of ABCG2 and ABCB6, their substrates are unlikely to be accommodated within the upper cavity when they are in the inward-facing state.

## The peptide binding site

To gain structural insight into the peptide-binding mechanism, we performed a single particle cryo-EM analysis of DM/CHS-purified ΔTMD0 in the presence of 9-mer peptide ([1]RRYQKSTEL[9])[2,5,24]. A dataset

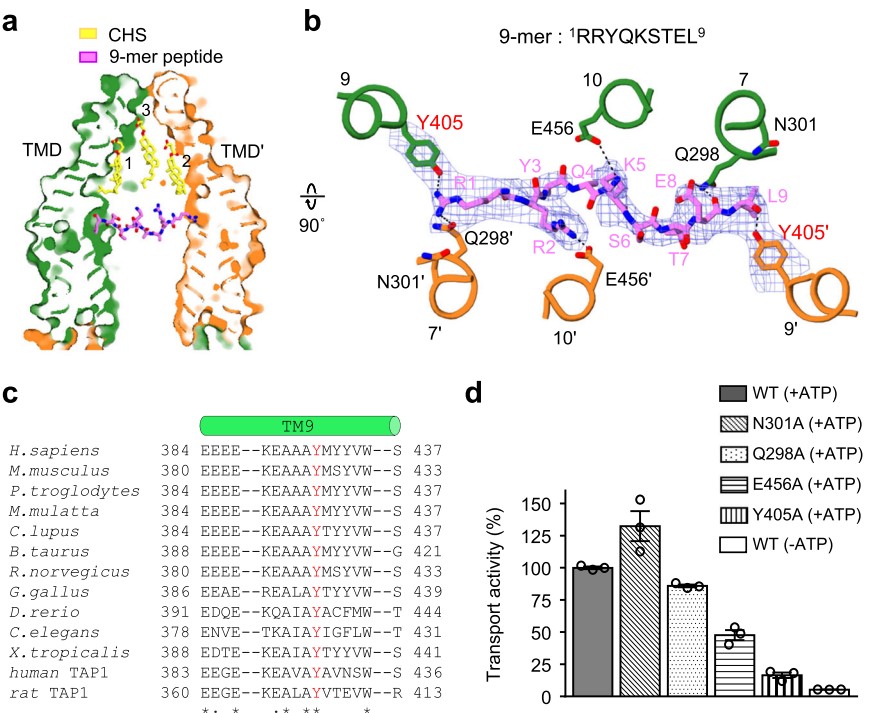

**Fig. 4 | Peptide binding site. a** Surface slab view of ΔTMD0 in the presence of 9-mer peptide and CHS. Bound peptide in the central cavity is depicted by the violet stick model and CHS in the upper cavity is represented by the yellow stick model. **b** Close-up view of the 9-mer peptide binding site. The residues interacting with bound peptides are shown in stick form. The EM density maps of the peptide and Y405 in each monomer are shown at the 4σ level. The tyrosine residue and glutamine residue at positions 3 and 4 (Y3 and Q4), respectively, of the peptide, were built as polyalanine due to the low electron density. **c** Alignment of the TM 9 sequences of TAPL homologs. The tyrosines highlighted in red indicate strictly conserved residues. **d** Peptide transport activities of the wild-type and ΔTMD0 mutants. Data for the wild-type in the presence of 5-IAF-labeled peptide and ATP was set as 100%. Values represent the means ± standard error of the mean (SEM) of triplicate measurements using the same batch of proteoliposomes. Source data is provided as source data file.

comprising 102,428 particles yielded a 3D reconstruction with an overall resolution of 4.0 Å. Although the micrograph quality of the DM/CHS-purified ΔTMD0 was similar to that of the nanodisc-reconstituted transporter, the final 3D reconstruction was less well-resolved. It seems likely that the dynamic and mobile nature of the detergent micelles renders the conformation of ΔTMD0 highly flexible. Nevertheless, the local resolution of the TM core region exceeded 3.7 Å resolution, allowing us to build a de novo model of the side chains in this region.

Our structure revealed an additional electron density, corresponding to the bound 9-mer peptide, extending horizontally across the central cavity (Fig. 4a). The peptide electron density faded towards the central region (Fig. 4b), probably due to peptide flexibility and/or the two-fold averaged map, preventing a detailed analysis of peptide geometry. Hence, in order to build the best possible structural model of the peptide, we maximized favorable local interactions with the protein and avoided steric collisions. The resulting structure revealed how the peptide is bound in TAPL. The arginine (R1) and the backbone carboxyl group of leucine (L9) at the peptide N- and C-termini, respectively, anchor to the transporter via hydrogen bonds with the strictly conserved Y405 of each monomer (Fig. 4b, c). The three positively charged residues (R1, R2, and K5) may enhance peptide binding through electrostatic interactions with neighboring residues Q298', E456', and E456, respectively. The side chain and the backbone carbonyl oxygen of the glutamic acid residue at position 8 (E8) form a salt bridge and a hydrogen bond with Q298, respectively. At the present resolution, it was not possible to identify further interactions responsible for stabilizing the extended peptide structure. Therefore, we do not exclude the possibility that ΔTMD0 can bind its peptide substrates in more than one way.

To confirm that the amino acids mentioned above are involved in peptide binding, we performed site-directed mutagenesis and transport assays in proteoliposomes using 5-IAF-labeled 9-mer peptide. N301 was chosen as a background control because it is close to Y405 but not involved in peptide binding (Fig. 4b). Mutations in the predicted protein–peptide contact sites reduced apparent peptide transport activity compared to the wild-type and N301A mutant (Fig. 4d). In particular, Y405A resulted in a complete loss of transport function and a 73-fold decrease in peptide affinity ($K_d$ of 51.9 nM vs. 3.8 µM) as determined by microscale thermophoresis (MST) (Fig. 5)[40], indicating that Y405 is a key molecular determinant that mediates high-affinity peptide binding to TAPL. The Y405A mutation was also associated with an apparent decrease in basal turnover and led to weak response to lipid substrates (Supplementary Fig. 8). Our results are entirely consistent with previous biochemical data. For example, peptide transport by human TAPL was blocked by modification (i.e. acetylation, amidation, or D-amino acid substitution) of the N- or C-terminus of 8-mer peptides[2]. Positively charged and hydrophobic residues at each end of the peptide are favored by TAPL, while negatively charged aspartate is disfavored[12]. This peptide binding specificity may be due to the fact that Y405 has a hydroxyl group on its aromatic ring (Fig. 4b).

## The lipid binding site

The structure of inward-facing, nanodisc-reconstituted ΔTMD0 in Fig. 3 contains an additional electron density within the upper cavity. The shape of the density clearly corresponds to a phospholipid molecule with two fatty acyl chains (Fig. 6a). The nanodiscs used in our structural studies were made up of the three major *E. coli* polar lipids, PE (67.0%), PG (23.2%) and cardiolipin (9.8%)[41]. However, as mentioned

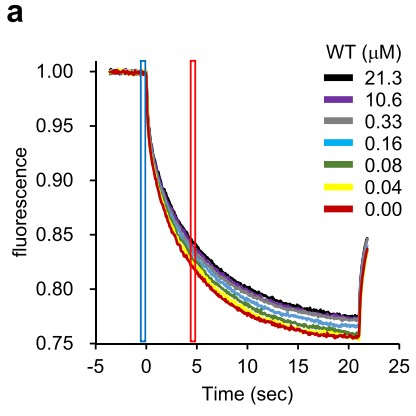

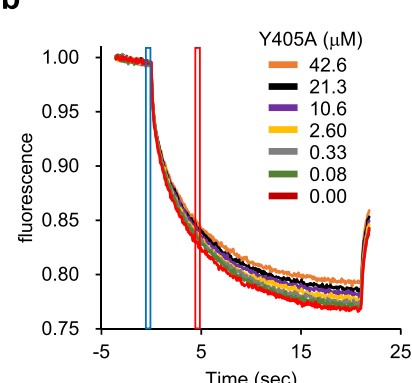

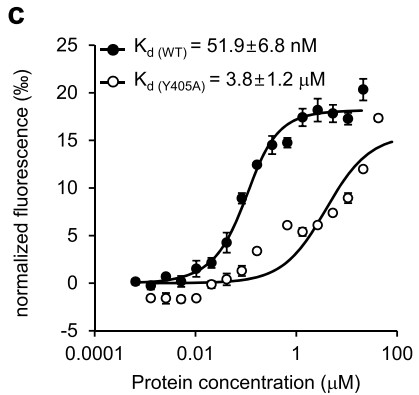

**Fig. 5 | Thermophoretic analysis of the ΔTMD0–peptide interaction. a** and **b** Microscale thermophoresis (MST) profiles of **a** the ΔTMD0- and **b** the Y405A-peptide (RRYQNSTC$^{Cy\text{-}5}$L) interaction. $F_{cold}$ (blue region) at 0 s represents the cooled state, and $F_{hot}$ (red region) at 5 s represents the heated state during thermophoresis. **c** Normalized fluorescence changes were measured to yield binding curves. Normalized fluorescence is defined as $F_{hot}/F_{cold}$. Each datum is the mean ± standard error of the mean (SEM) of at least three independent experiments using the same batch of DM-purified protein. Source data is provided as source data file.

above, no PE-stimulated ATPase activity was detected due to non-specific aggregation of PE in the buffer. Also, cardiolipins with four acyl chains cannot fit on the map. We, therefore, built an atomic model of the PG (16:0–18:0) molecule in the head-up direction as a phospholipid model (Fig. 6b).

As seen in Fig. 6a, the PG molecule is located around the two-fold axis of the ΔTMD0 dimer, where it generates a large and symmetric electron density in the head group region, but relatively weak densities in the two lipid chains (Fig. 6b), possibly due to a molecular averaging effect, the flexibility of the lipid chains or multiple binding modes. The bound PG mainly interacts through hydrophobic interactions between the lipid acyl chains and non-polar residues of both monomers, including L201, V227, L424, I442, I445, I446, F449, and V450 (Fig. 6c). Because of the lipid promiscuity of ΔTMD0 (Fig. 1c–e), it is possible that some endogenous lipids, such as PS or PC, co-purified with ΔTMD0 and occupied the upper cavity of the transporter without causing a severe steric collision (Supplementary Figs. 9 and 10).

Similarly, ΔTMD0 purified in DM/CHS micelles contained extra electron densities at a position equivalent to the binding site of the PG molecule (Figs. 4a and 6d). The shape of the density resembled that of sterol-type lipids (Supplementary Fig. 11a). As the CHS was separated from the purified protein by TLC (Supplementary Fig. 11b), we fitted three CHS molecules (CHS1–3) into additional EM density maps. CHS1 and CHS2 are close to each monomer and a central CHS3 is positioned around the two-fold axis of the ΔTMD0 dimer interface (Fig. 6d). The CHSs are positioned vertically in the upper cavity where the polar hemisuccinate moieties project toward the apex of the transport

pathway, occupying almost the entire upper cavity. Because the CHS-bound structure could only be solved at a rather low resolution, it was impossible to identify the specific residues essential for CHS binding. Nonetheless, we observed that all bound CHS molecules were surrounded mainly by hydrophobic residues within a 5 Å of distance (L201 and T205 from TM 5, L416 and L424 from TM 9, F449, I445, I446, and V450 from TM 10) (Fig. 6e). Taken together, our findings suggest that the hydrophobic nature of the upper cavity is critical for stabilization and/or active transport of bound lipids.

## A lateral gate for lipid entry

The inward-facing ΔTMD0 structure has a wide lateral gate facing the cytosolic leaflet of the lysosomal membrane, with an opening of ~18 Å in the largest diameter (Fig. 3d). Intriguingly, a density corresponding to a detergent molecule (or lipid) is present near the lateral opening of ΔTMD0 (Fig. 6f). It is surrounded by glycine-containing TM 8 and 10 (G343 on TM 8 and G452, G459, G464 and G468 on TM 10), with the acyl chain of the detergent molecule (or lipid) exposed to the micelles, while the polar head group is intercalated into the transport pathway. Similar structural features were observed in the *Shigella flexneri* lipopolysaccharide transporter LptB$_2$FGC[42] and the *C. elegans* multidrug transporter P-glycoprotein (which also functions as a lipid translocase)[36,43]. These findings suggest a possible lipid entry mechanism for TAPL, in which lipid substrates enter the TM cavity through the lateral opening, while the spatially clustered glycines confer enough flexibility to permit the conformational changes associated with lipid binding and/or entry[44,45].

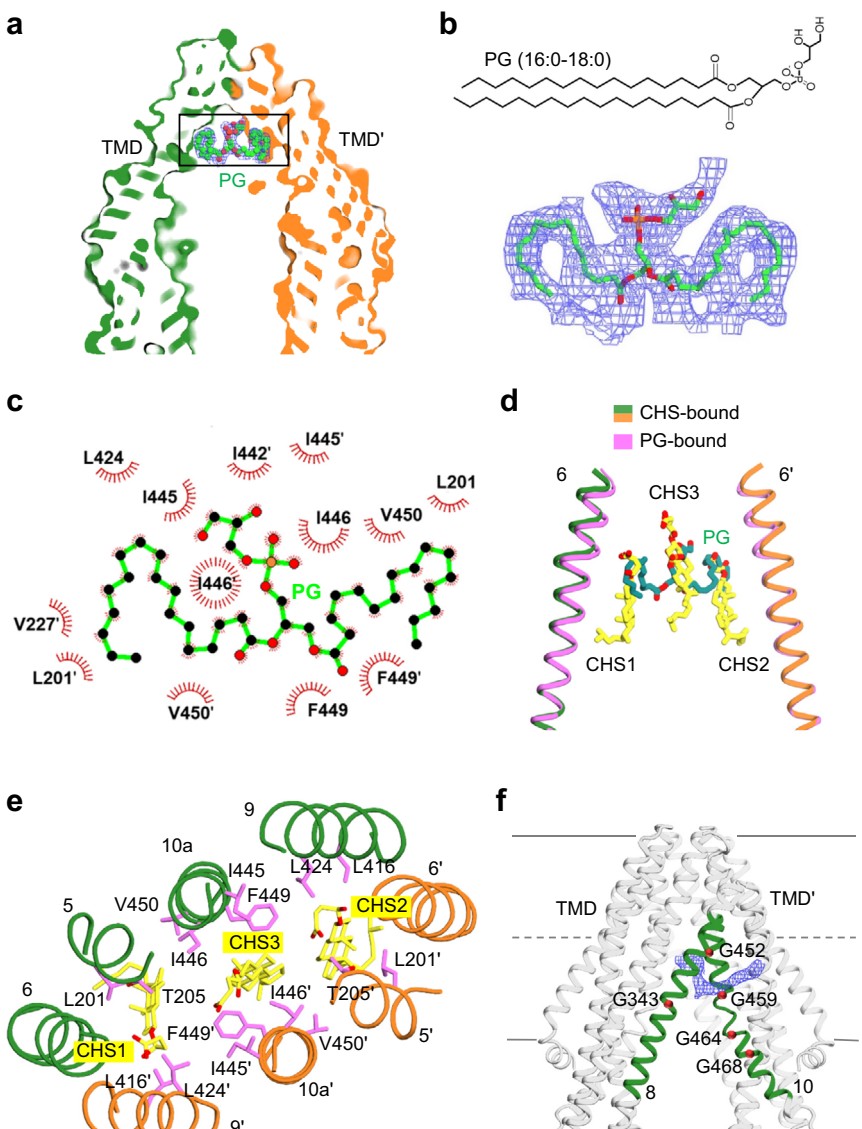

**Fig. 6 | Lipid binding site. a** Surface slab view of inward-facing ΔTMD0 in nanodisc. A PG (16:0–18:0) molecule bound in the upper cavity is represented by a space-filling model. The EM density of the bound PG is shown at 3.5σ level. Carbon, oxygen, and phosphorus atoms are colored green, red, and orange, respectively. Zoom-in view of the boxed region is presented in (**b**). **b** The chemical structure (top) and EM density (3σ level, bottom) of bound PG. **c** Interactions between ΔTMD0 and PG were analyzed with LigPlot+ software. The PG (green) is shown in ball and stick representation. Residues involved in nonpolar and van der Waals interactions within a distance of 4.5 Å are depicted as red semicircles. **d** Structural superimposition of PG and CHS binding sites. Bound PG and CHSs are represented by the green and yellow stick models, respectively. **e** Top view of the CHS binding site in the upper cavity. Bound CHSs and nearby residues are shown as sticks. **f** EM density of detergent molecule (or lipid) inserted into the lateral gate of DM/CHS-purified ΔTMD0 (4σ level). TM 8 and TM 10 are highlighted in green. The glycine residues adjacent to the lateral gate are shown as red spheres.

## Structural comparison of TAPL with other TAP family members

Although the human TAP1/2 complex and its bacterial homolog, TmrAB, are functionally very similar to TAPL, there are notable structural differences between their inward-facing conformations. For example, in TAP1/2 there is a global outward shift of the TM helices at the apex, although its NBD separation is similar to that of ΔTMD0 (Supplementary Fig. 12a–c)[19,20]. This may be caused by the binding of herpes simplex virus protein ICP47, which has a long hairpin-like structure that plugs the translocation pathway vertically and expands the TM cavity. Unlike with ΔTMD0, no lateral entry was observed with the TAP1/2 complex, and sequence comparison revealed that the F449 residue of TAPL, which separates the large central cavity from the upper cavity, is not conserved in TAP1 or TAP2 (M452 and E417, respectively) (Supplementary Fig. 12d). Of the amino acids critical for peptide recognition, only Y405 and E456 of TAPL are strictly

conserved in TAP1 (Y408 and E459, respectively). Also, of the glycine clusters surrounding the lateral gate, only G343 (in TAP1) and G452 (in TAP2) are conserved, at positions 346 and 420, respectively.

In contrast to the situation in the TAP1/2 complex, the TM helices of TmrAB are shifted inwards compared to those of ΔTMD0, giving rise to a relatively small TM cavity and more closely spaced NBDs (Supplementary Fig. 13)[21]. Note that TmrAB contains only one lateral opening (in TmrB, but not in TmrA). A recent mass spectrometry analysis indicated that various lipid A species bind to TmrAB and dissociate upon ATP hydrolysis[46], suggesting that TmrAB may function as a lipid A exporter.

## The nucleotide-bound outward-facing conformation

To obtain a nucleotide-bound structure, nanodisc-reconstituted ΔTMD0 was mixed with 9-mer peptide and Mg²⁺/ADP·BeF₃

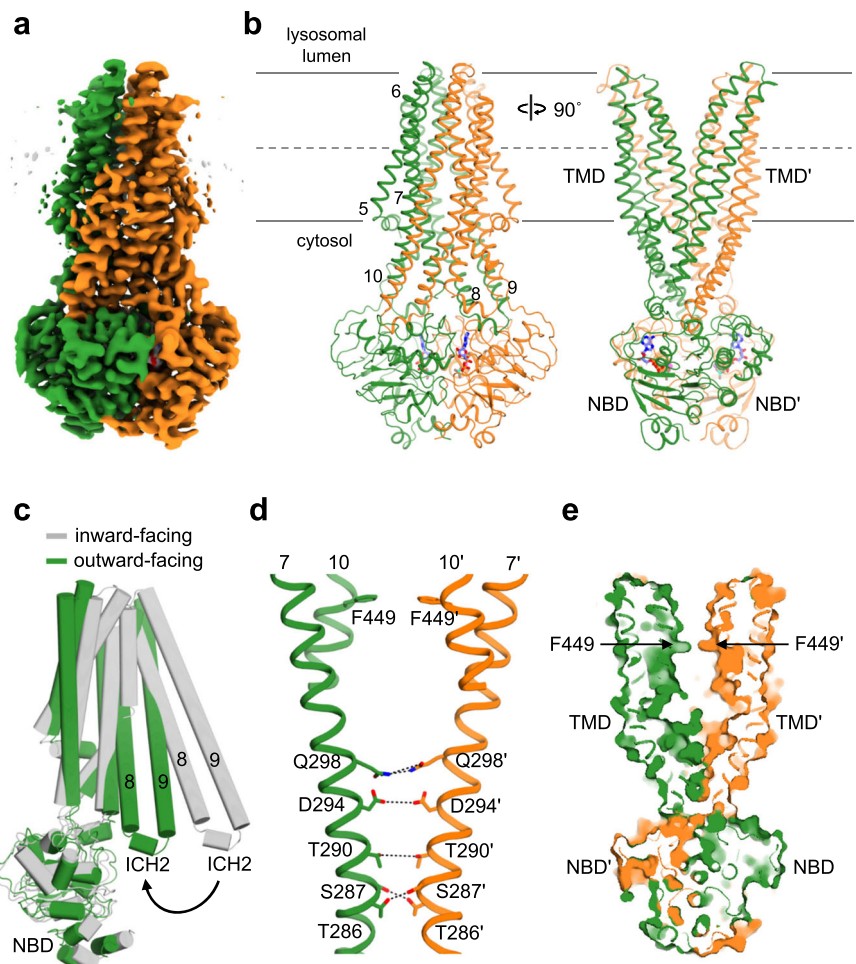

**Fig. 7 | The outward-facing conformation. a** 3D reconstruction of outward-facing ΔTMD0. **b** Cartoon representation of outward-facing ΔTMD0. The bound $Mg^{2+}$ and ADP·BeF$_3$ are represented by a sphere and a stick, respectively. **c** Structural comparison of the inward-facing and outward-facing monomer. Both are from nanodisc-reconstituted ΔTMD0. **d** The residues of TM 7 involved in the stabilization of the outward-facing conformation are shown as sticks. The F449 residue on TM 10 is also shown. **e** Surface slab view of outward-facing conformation.

(mimicking the pre-hydrolysis state) before preparing cryo-EM grids[47]. The final map has a resolution of 3.2 Å, allowing accurate modeling of the $Mg^{2+}$/ADP·BeF$_3$ complex and side chain conformations (Fig. 7a). The structure represents the typical outward-facing shape of ABC transporters, in which the two NBDs come close together and the TM helices have a very wide opening towards the lysosomal lumen (~36 Å in the longest diameter), while they are tightly packed on the cytosolic side (Fig. 7b, c). As a consequence, ΔTMD0 forms a V-shaped substrate transport pathway open not only to the lysosomal lumen but also laterally to the membrane luminal leaflet. We did not observe any density corresponding to lipids or peptides, indicating that both substrates are exported from ΔTMD0 prior to ATP hydrolysis. In the dimer interface, T286, S287, T290, D294, and Q298 on TM 7 interact electrostatically to stabilize the outward-facing conformation (Fig. 7d). Interestingly, the F449 residues remain protruding towards the transport pathway and pack closely together to create a latch-like architecture (Fig. 7e). Consequently, the central cavity is completely isolated from solvent, thus probably preventing the expelled substrate from re-entering.

Structure comparison indicates that the large-scale conformational change from the inward- to the outward-facing state repositions key residues involved in substrate binding, and reorganizes the TM cavities. As a result, the hydrophobic residues that interact with the lipids in the upper cavity are moved away from the binding site and directed towards the hydrophobic interior of the TMD region

(or membrane) (Supplementary Fig. 14a, b). In addition, the volume of the central cavity decreases, reducing the distance between the Y405 residues from 30 to 20 Å (Supplementary Fig. 14c). This is too small to optimally bind a 9-mer peptide in the extended configuration. These combined structural rearrangements may be unfavorable for accommodating substrates, thus facilitating their release by reducing binding affinities.

In the NBD region, the two ADP·BeF$_3$ molecules are sandwiched between the two NBDs (Fig. 8a). The α- and β-phosphates of ADP and BeF$_3$ interact with the Walker A motif and Q-loop from one NBD and the ABC signature motif of the other NBD (Fig. 8b). The adenine ring of ADP forms a π–π interaction with the stacking aromatic residue Y509 of the A-loop. The 2′ and 3′ hydroxyls of the ribose moiety form hydrogen bonds with Q643 of the ABC signature motif. The $Mg^{2+}$ ion is coordinated by the β-phosphate of ADP, BeF$_3$, S542 of the Walker A motif, and Q583 of the Q-loop. We also found that D274 of the intracellular coupling helix 1 (ICH1) extends toward the NBD dimer interface to interact with Y552 of one NBD, Q638 of the other, and the amino group of the C6 carbon of ADP (Fig. 8b). These electrostatic interactions likely serve to further stabilize the NBD dimer.

## Discussion
In this work, we have investigated the ability of peptide and lipid molecules to affect the ATPase and transport activity of mouse TAPL. Consistent with previous data from human TAPL[2,5,24], the results

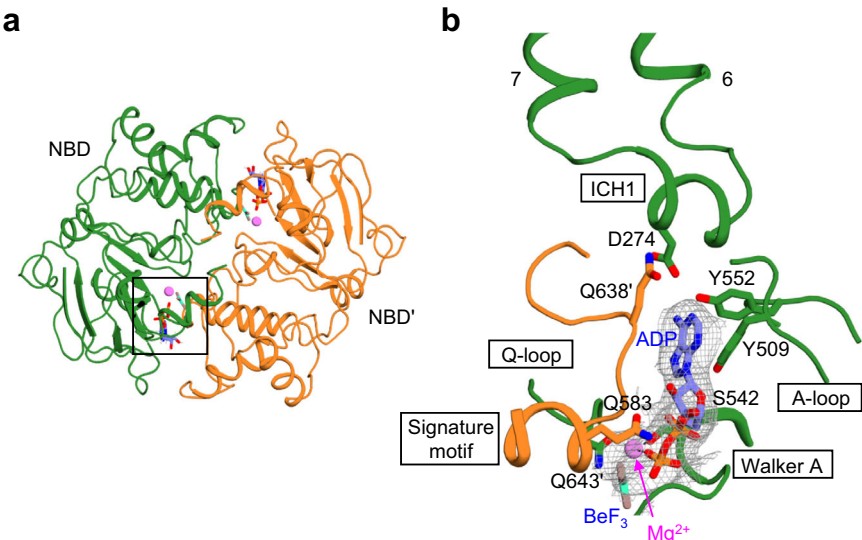

**Fig. 8 | The nucleotide-binding site. a** The NBD dimer is viewed from the cytoplasm. Bound $Mg^{2+}$ and ADP·$BeF_3$ are represented by sphere and stick, respectively. The area enlarged in (**b**) is boxed. **b** Close-up view of the ADP·$BeF_3$ binding site. The bound $Mg^{2+}$ is represented by a magenta sphere. The EM density of ADP·$BeF_3$ is shown at the $4\sigma$ level.

indicate that ΔTMD0 preferentially binds and transports 9-mer peptides, but that peptides do not accelerate their ATP hydrolytic activity (Fig. 1a, b). We also obtained the direct measurement of an ATP-dependent floppase activity of ΔTMD0 using the fluorescent lipid analog, NBD*-PS, in a reconstituted system (Fig. 2). Unfortunately, no translocation of other lipid species could be detected due to technical limitations of our model system (i.e. competition between unlabeled natural lipid and NBD*-lipids). Nevertheless, our structures and biochemical data suggest that ΔTMD0 translocates various lipids in the membrane (Figs. 1c–e, 4a, and 6a). This TAPL-mediated lipid redistribution suggests that TAPL is involved in regulating lipid trafficking between the inner and outer leaflets of lysosomal membranes, or between other organelles or intraluminal vesicles, and thus may contribute to the characteristic lipid composition of the lysosome[48,49].

The structures obtained in this study provide valuable insights into the functional properties of TAPL, which are otherwise difficult to explain. First, many structures of ABC transporters have shown that the binding of substrate induces a global inward movement of the TM helices, leading to an approximation of the two NBDs[37,50]. However, the structural comparison indicates that peptide binding to ΔTMD0 has little if any impact on the overall inward-facing structure (Cα r.m.s.d. of 1.4 Å) and the degree of NBD separation (Supplementary Fig. 15). We think that the insertion of the peptide parallel to the membrane plane may prevent the NBDs from moving closer so that the transporter is fixed in the inward-facing open state and the catalytic cycle is not accelerated by the presence of a peptide (Figs. 1b, 4a)[24]. Second, our structure reveals that both termini of the peptide are held in place by strategically placed Y405 residues, while the central parts form only minor contacts with the protein (Fig. 4b). The nature of this binding explains why TAPL possesses an ability to engage peptides that vary greatly in the size and sequence of their central regions[2,12,13,51]. Third, we can now explain the peptide length preference of TAPL. In our structure, the distance between the primary anchor residue Y405 in the two monomers is ~30 Å, thus allowing a peptide of 8–10 amino acids to adopt an extended conformation spanning the central cavity (Fig. 4a, b, and Supplementary Fig. 14c). Hence, peptides consisting of <8 amino acids may be too short to bridge the binding sites even in the fully extended conformation. By contrast, long peptide chains exceeding 10 amino acids may bulge in the middle or make sharp turns at specific positions[52], which are thought to inhibit proper transport cycling[24].

It is noteworthy that the peptide binding modes of the TAPL and TAP complex are very similar. For example, fluorescence cross-correlation spectroscopy showed that the TAP complex binds only one peptide at a time[53], and structure-based sequence alignment revealed that rat TAP1 has a tyrosine residue at position 385 (corresponding to Y408 in human TAP1) that is equivalent to Y405 in ΔTMD0 and is crucial for peptide binding (Fig. 4c)[54]. Also, screening of a combinatorial peptide library demonstrated that the TAP complex has a narrower peptide spectrum than TAPL[2,11–13], but that both transporters have a strong preference for 9-mer peptides with positively charged, hydrophobic, or L-amino acids at either end[51,55–59]. Furthermore, electron paramagnetic resonance (EPR) spectroscopy has revealed that the mobility of the peptide bound to the TAP complex is greatly reduced at either end[52], and a molecular docking study has suggested that it adopts an extended conformation parallel to the membrane plane[60].

In summary, we have revealed molecular details of the two distinct transport mechanisms of TAPL. These have significant implications for understanding how structurally and chemically different substrates (i.e. peptides vs. lipids) are bound and translocated by a single transporter. In the resting state, the transporter may adopt a conformation similar to that observed in lipid-bound TAPL or it may have a more open inward-facing conformation that allows the substrate to readily access the binding site (Fig. 9). Peptides gain access through the cytoplasmic opening into the central cavity, whereas lipids enter the TM cavity through a lateral opening of the cytosolic membrane leaflet and are then flipped to the luminal leaflet. As previously observed for other ABC transporters[22,61–63], the binding of substrate and ATP promotes the formation of the outward-facing conformation (pre-hydrolysis state) in which TAPL exports the bound peptide directly into the lysosomal lumen for degradation, or releases the lipid to the membrane luminal leaflet, from which it diffuses away, as in the original "flopping model" of Higgins and Gottesman[64]. ATP hydrolysis followed by phosphate release (post-hydrolysis state) then resets the transporter back to the resting state.

## Methods
### Cloning, expression and purification
The genes encoding full-length mouse TAPL (Genebank accession code AB045382.2) and its truncation mutant devoid of the N-terminal 142 residues (ΔTMD0) were cloned between the BamHI and NotI sites in the pVL1393 baculovirus transfer vector (BD

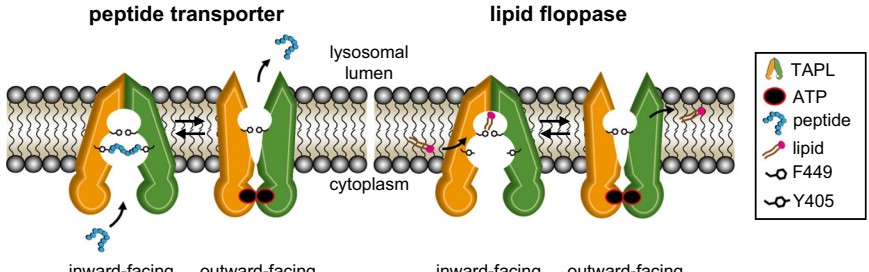

**Fig. 9 | TAPL plays a dual role as a peptide transporter and lipid floppase.** In the resting state, TAPL probably has a conformation similar to that observed for lipid-bound TAPL or it may have a more open, inward-facing conformation for good access of substrates to their binding sites. Peptides can gain access to the central cavity via the cytosolic opening, whereas lipids enter the hydrophobic upper cavity through the lateral gate. The binding of substrate and ATP triggers a change to the outward-facing conformation in which TAPL releases the bound peptide into the lysosomal lumen and/or the lipid into the membrane luminal leaflet. ATP hydrolysis resets the transporter to the resting state for another cycle. The TMD0 is omitted for simplicity.

Biosciences). A thrombin cleavage site was introduced between the TAPL gene and the C-terminal enhanced green fluorescence protein (eGFP) and the decahistidine (10XHis) tag. Mutations were introduced by polymerase chain reaction (PCR)-based site-directed mutagenesis (Table S1). Successful mutations were verified by DNA sequencing. The recombinant plasmids were transfected into *Spodoptera frugiperda* (Sf9, Expression Systems) cells using Cellfectin II transfection reagent (Gibco) and BestBac 2.0 linearized baculovirus DNA (Expression Systems). All proteins were expressed in High Five (Hi5, Expression systems) cells for 72 h at 28 °C. Protein expression was monitored by fluorescence microscopy.

The transgenic cells were harvested at $10,000 \times g$ for 15 min and resuspended in lysis buffer containing 20 mM HEPES–NaOH pH 7.5, 200 mM NaCl, 10% (v/v) glycerol, 10 mM $MgCl_2$, 40 µg/mL DNase I (GoldBio) and 1 mM phenylmethylsulfonyl fluoride (PMSF, GoldBio). After sonication at 40% amplitude with 5 s ON/OFF pulses for 6 min (Branson sonifier, 3.2 mm tip), the membranes were collected by centrifugation at $300,000 \times g$ for 1 h. Proteins were solubilized for 2 h with 2% (w/v) n-dodecyl-β-D-maltopyranoside (DDM, Anatrace) with and without 0.2% (w/v) cholesteryl hemisuccinate (CHS, Anatrace). Insoluble cell debris was removed by centrifugation at $300,000 \times g$ for 30 min and the supernatant was purified by anti-GFP DARPin-conjugated agarose chromatography[65]. After washing the resins with a buffer containing 20 mM HEPES–NaOH pH 7.5, 200 mM NaCl, 0.17% (w/v) n-decyl-β-D-maltopyranoside (DM, Anatrace) with and without 0.017% (w/v) CHS, bound proteins were eluted by on-column thrombin (Lee Biosolutions) cleavage. The eluted proteins were further purified by gel filtration chromatography using Superdex 200 Increase 10/300 GL column (GE Healthcare). The pooled fractions were stored at −80 °C until use. All purification steps were performed on ice or at 4 °C.

### Nanodisc preparation
The synthetic gene for membrane scaffold protein 1 (MSP1), a deletion mutant devoid of the N-terminal 43 residues of human apoA-I, was prepared as previously described[66]. Briefly, MSP1 was cloned into pET21a vector (Novagen) with a C-terminal hexahistidine (6XHis) tag. The protein was produced in *Escherichia coli* BL21(DE3) cells cultured in Luria-Bertani medium at 37 °C[67]. At $OD_{600}$ 0.7, protein expression was induced with 1 mM isopropyl-β-D-thiogalactopyranoside (IPTG, GoldBio) for 5 h at 30 °C. MSP1 was purified by Ni-NTA affinity chromatography, HiTrap Q (GE Healthcare) anion exchange, and Superdex 200 Increase 10/300 GL gel filtration chromatography.

The lipids (Avanti Polar Lipids, Inc., purity > 99%) used for nanodisc assembly were *E. coli* polar lipid extract, 1-palmitoyl-2-oleoyl-sn-glycero-3-phospho-l-serine (POPS), 1-palmitoyl-2-oleoyl-sn-glycero-3-phospho-(1′-rac-glycerol) (POPG), and 1-palmitoyl-2-oleoyl-glycero-3-phosphocholine (POPC). Each lipid was dissolved in 0.17% (w/v) DM and 0.017% (w/v) CHS and mixed with MSP1 and DM/CHS-purified

ΔTMD0 at a molar ratio of 150:4:1. After incubation on ice for 10 min, the mixture was treated with 60 mg of Bio-beads (Bio-Rad) at 4 °C for 4 h to remove detergent. The supernatant was additionally incubated overnight with 120 mg of fresh Bio-Beads at 4 °C. The final supernatant was injected onto a Superdex 200 Increase 10/300 GL column equilibrated with a buffer of 20 mM HEPES–NaOH pH 7.5 and 200 mM NaCl. The elution fractions containing protein-loaded nanodiscs were pooled and used for cryo-EM and functional analyses.

### Peptide synthesis and fluorophore labeling
All peptides used in this study were synthesized by the standard solid phase technique (ANYGEN Co., Ltd.). 5-Iodoacetamido fluorescein (5-IAF)-labeled peptides were prepared as previously described[68]. Briefly, peptides containing a single cysteine residue were incubated with 5-IAF (Sigma-Aldrich) at a 1:2 molar ratio in 100 mM HEPES–NaOH pH 7.5 for 2 h at RT. The 5-IAF-labeled peptide was separated from the native peptide and excess 5-IAF by Shim-pack PREP-ODS column-equipped reversed-phase high-performance liquid chromatography (Shimadzu 10Avp). The extent of 5-IAF labeling was confirmed by mass spectrometry.

### Peptide transport assay
Proteoliposomes for peptide transport assays were prepared according to the protocol of Zhao et al., with some modifications[12]. Briefly, *E. coli* polar lipid extract dissolved in chloroform was dried under a gentle stream of nitrogen. The thin film of lipids was placed under a desiccator overnight to completely remove residual chloroform and then hydrated in 1X PBS buffer pH 7.4 (137 mM NaCl, 2.7 mM KCl, 8 mM $Na_2HPO_4$, and 2 mM $KH_2PO_4$) with 10 mM $MgCl_2$ and 5% (v/v) glycerol. After water bath sonication at 22 °C for 5 min (Powersonic 610, KLEENTEK), the lipid solution was extruded using a Mini-Extruder equipped with a polycarbonate membrane filter (200 nm pore) (Avanti Polar Lipids, Inc.) to obtain unilamellar vesicles, which were destabilized with 0.36% (w/v) DM. The DM-purified protein and lipid suspension were mixed in a 1:50 (w/w) ratio. After incubation at 4 °C for 2 h, the mixture was treated with Bio-Beads SM-2 resin (60 and 120 mg, see above) at 4 °C for a total of 18 h to remove detergent. The proteoliposomes were collected by centrifugation at $200,000 \times g$ for 1.5 h at 4 °C and resuspended in the buffer described above.

The peptide transport activity of ΔTMD0 was measured as described previously[68]. Briefly, 50 µL of ΔTMD0-loaded proteoliposomes (~0.6 µg protein) were added into 200 µL reaction buffer containing 1X PBS pH 7.4, 3 mM ATP, and 10 mM $MgCl_2$ in the presence or absence of 10 µM 5-IAF-labeled peptides. To investigate transport activity, the mixture was incubated at 37 °C for 10 min. The reaction was terminated by incubating the mixture for 20 min on ice. Proteoliposomes were collected by centrifugation at $200,000 \times g$ for 30 min and then washed twice with 1X PBS. The proteoliposomes were

completely disrupted by adding 10% (w/v) SDS solution (final concentration, 2%). The transported peptide was measured using a microplate reader (Gemini XPS) with excitation at 485 nm and emission at 520 nm.

## ATP hydrolysis assay

The ATPase activities of proteins obtained after various purification procedures were determined by NADH-coupled spectrophotometric assays[69]. 10 μg of DM-purified or nanodisc-reconstituted protein was added to 140 μL of reaction buffer containing 50 mM HEPES–KOH pH 8.0, 60 μg/mL pyruvate kinase (Roche), 16 μg/mL lactate dehydrogenase (Roche), 10 mM phosphoenolpyruvate (Roche), 0.3 mM NADH, 3 mM ATP, 10 mM MgCl$_2$, and 0.17% (w/v) DM in the presence or absence of 250 μM peptide. For proteins reconstituted into nanodiscs, DM was not added to the reaction buffer. For lipid-stimulated ATPase assays, the thin film of lipids was hydrated in 100 μL of buffer containing 50 mM HEPES–KOH pH 8.0 and 10 mM MgCl$_2$ to yield a 10 mM solution. After sonication of the lipid solution for 10 min to decrease its turbidity, the reaction was initiated by adding varying concentrations of lipids to the mixture. ATP hydrolysis activity was monitored at 37 °C for 3 min by measuring the rate of decline of NADH absorbance at 340 nm. To inhibit ATP hydrolysis, the reaction mixture was incubated with 3 mM AMP-PNP or 8 mM NaF/2 mM BeSO$_4$ for 5 min at RT. The kinetic parameters of ATP hydrolysis were calculated by fitting the initial rates of ATP hydrolysis to the Michaelis–Menten equation using PRISM 4.0 (GraphPad software).

## Flip-flop assay of NBD*-labeled lipids

All lipids used in reconstitution were purchased from Avanti Polar Lipids, Inc. (purity > 99%). The in vitro flip–flop assay was performed using NBD* dye-labeled proteoliposomes as described previously[70]. Briefly, E. coli polar lipids and egg PC were mixed in a 1:1 molar ratio, and 0.5% (mol/mol) NBD*-lipid was added. The lipid mixture was then dried under nitrogen gas. The NBD*-lipids used were 18:1-06:0 NBD*-PS (1-oleoyl-2-{6-[(7-nitro-2-1,3-benzoxadiazol-4-yl)amino]hexanoyl}-sn-glycero-3-phosphoserine), 18:1-06:0 NBD*-PC (1-oleoyl-2-{6-[(7-nitro-2-1,3-benzoxadiazol-4-yl)amino]hexanoyl}-sn-glycero-3-phosphocholine), 18:1-06:0 NBD*-PG (1-oleoyl-2-{6-[(7-nitro-2-1,3-benzoxadiazol-4-yl)amino]hexanoyl}-sn-glycero-3-[phospho-rac-(1-glycerol)]), 18:1-06:0 NBD*-PE (1-oleoyl-2-{6-[(7-nitro-2-1,3-benzoxadiazol-4-yl)amino]hexanoyl}-sn-glycero-3-phosphoethanolamine), and 25-NBD*-Cholesterol (25-[N-[(7-nitro-2-1,3-benzoxadiazol-4-yl)methyl]amino]−27-norcholesterol). Thin lipid films were solubilized in 20 mM HEPES–NaOH pH 8.0, 200 mM NaCl and 5% (v/v) glycerol, incubated at RT for 30 min and subjected to 5 freeze–thaw cycles. After extrusion with a Mini-Extruder at 65 °C, final lipid concentrations were determined by phosphorus assays[71]. Briefly, 10 μL of extruded lipid was added to 10 μL of 2% (w/v) ammonium molybdate and 300 μL of concentrated perchloric acid. The mixture was heated at 250 °C for 30 min, then cooled to RT and mixed with 1.5 mL of 0.4% (w/v) ammonium molybdate and 250 μL of 9% (w/v) L-ascorbic acid. It was again heated at 100 °C for 6 min and cooled to RT. Lipid concentrations were calculated from the absorbance at 820 nm using a standard curve prepared with NaH$_2$PO$_4$ solution. After accurately determining the lipid concentration, DM-purified protein and 5 mM lipid suspension (solubilized in 5 mM DM) were mixed 1:25 (w/w). After incubation at 4 °C for 1.5 h, DM detergent was removed with 75 mg Bio-beads SM-2 by periodic gentle shaking for 1 h at 4 °C (Supplementary Fig. 5a). The beads were replaced once, followed by an additional incubation at 4 °C for 2 h to completely remove detergent. The resulting proteoliposomes were collected by ultracentrifugation at 200,000×g for 1.5 h at 4 °C.

To measure the flip–flop activity, 10 μL of 2 mM NBD*-labeled proteoliposomes were added to 100 μL of reaction buffer (20 mM HEPES–NaOH pH 8.0, 200 mM NaCl, 5 mM ATP and 5% (v/v) glycerol) in the presence of 2 mM MgCl$_2$ (or NaCl), and incubated at 37 °C for

1 h. The mixture was transferred to a 96-well plate and time-dependent fluorescence ($\lambda_{ex}/\lambda_{em}$ = 460/538 nm) was recorded every 30 s for 26 min using a fluorescence spectrophotometer (Gemini XPS). After the baseline fluorescence had stabilized, 10 mM sodium dithionite was added to quench the fluorescence of NBD*-lipid present in the outer leaflets of the liposomes. The initial rapid drop and subsequent slow decline in fluorescence intensity were monitored. When a new steady state was established, 1% (v/v) Triton X-100 was added to completely disrupt the liposomes. The net percentage of NBD*-lipid translocated across the lipid bilayer was calculated from the following equation[72]: the percentage of protected NBD*-lipid in the inner leaflet = [($F_D$−$F_0$)/($F_T$−$F_0$)]×100, and that of accessible NBD*-lipid in the outer leaflet = [($F_T$−$F_D$)/($F_T$−$F_0$)]×100, where $F_T$ is the total fluorescence of the sample before the addition of dithionite, $F_D$ is the fluorescence of the sample after quenching with dithionite, and $F_0$ is any residual fluorescence of the sample after solubilization with Triton X-100. Fluorescence values ($F_T$, $F_D$, $F_0$) were calculated as the averages of 5 points on the respective plateau lines. The ΔTMD0-mediated lipid floppase activity was determined by subtracting the percentage of NBD*-lipid fluorescence measured in the presence of Na$^+$/ATP from that measured in the presence of Mg$^{2+}$/ATP.

## Quantification of detergent

Detergent removal using Bio-Beads SM-2 was monitored colorimetrically, as previously described[33]. Briefly, a 500 μL solution containing 10 mM DM and 20 mM HEPES–NaOH pH 7.5 was incubated with different amounts of Bio-Beads for 1 h at 4 °C with gentle shaking. In parallel, 500 μL liposomes composed of E. coli polar lipids and egg PC in a ratio of 1:1 (mol/mol) were solubilized with 3.6 mM DM, and were incubated with Bio-beads as described above. Then, 50 μL aliquots of the mixtures were added to 250 μL of 5% (v/v) phenol and 600 μL of concentrated sulfuric acid (95–98%). After cooling to RT, 200 μL aliquots of the mixtures were transferred to a 96-well plate, and absorbance was measured at 490 nm. Detergent concentrations were calculated using a DM standard curve.

## Microscale thermophoresis assay

Binding of 9-mer peptide (RRYQNSTC$^{Cy-5}$L) to DM-purified ΔTMD0 was monitored by microscale thermophoresis as described previously[40]. Briefly, 9-mer peptide was incubated with a 10x molar excess of Cy5 dye (Thermo Fisher Scientific) in 50 mM HEPES–NaOH pH 7.5 at RT for 2 h. The labeled peptide was purified from the excess dye with a desalting column. Cy5-labeled peptide (0.2 μM) was added to serial 2-fold dilutions of protein. The mixtures were incubated for 10 min at RT and loaded onto Monolith NT.115 standard-treated capillaries (NanoTemper Technologies). Thermophoretic movements were induced by infrared laser activation. Subsequent fluorescence changes were monitored with a Monolith NT.115 pico device and plotted against protein concentration. The results were analyzed with MO.Affinity Analysis software.

## Thin layer chromatography (TLC)

DM/CHS- or DM-solubilized protein was purified by gel filtration chromatography equilibrated with a buffer containing 20 mM HEPES–NaOH pH 7.5, 200 mM NaCl and 0.17% (w/v) DM. Protein-bound lipids were extracted in a mixture of chloroform and methanol (1:2, v/v). The lower chloroform phase was collected and dried under a gentle stream of nitrogen gas. The dried lipid film was re-dissolved in 30 μL of chloroform/methanol (1:2, v/v). The extracts were loaded on a TLC Silica gel 60 RP-18 F254s plate (Merck) and developed in a solvent mixture of chloroform/methanol/ammonia/water (5:3:0.3:0.15, v/v/v/v)[73]. The plate was dried and stained with potassium permanganate solution (1.5 g of KMnO$_4$, 10 g of K$_2$CO$_3$, and 1.25 mL of 10% NaOH (w/v) in 200 mL H$_2$O) to visualize lipids.

We also used TLC to determine if the phospholipids used in this study were contaminated with lyso-phospholipids or other impurities. The lyso-phospholipids used as controls were 16:0 Lyso-PG (1-palmitoyl-2-hydroxy-sn-glycero-3-phospho-(1'-rac-glycerol)), 16:0 Lyso-PC (1-palmitoyl-2-hydroxy-sn-glycero-3-phosphocholine), 16:0 Lyso-PE (1-palmitoyl-2-hydroxy-sn-glycero-3-phosphoethanolamine), and 17:1 Lyso-PS (1-(10Z-heptadecenoyl)−2-hydroxy-sn-glycero-3-[phospho-L-serine]) (Avanti Polar Lipids, Inc., purity > 99%; may contain up to 10% 2-Lyso phospholipid isomer).

### Cryo-EM sample preparation and data collection

Aliquots of 3 µL of purified protein were applied to glow-discharged 300-mesh Au R1.2/1.3 Holey carbon grids (Quantifoil). For the peptide-bound form, 1 mM 9-mer peptide ([1]RRYQKSTEL[9]) was added to DM/CHS-purified ΔTMD0 and incubated on ice for 30 min. For the ADP·BeF$_3$-bound form, the protein reconstituted into the nanodiscs formed by *E. coli* polar lipids was incubated with 1 mM 9-mer peptide, 2 mM ATP, 2 mM MgCl$_2$, 8 mM NaF, and 2 mM BeSO$_4$ at 37 °C for 10 min. The grids were blotted for 1–2 s with a blot force of 3–5 and plunge-frozen in liquid ethane cooled with liquid nitrogen using a Vitrobot Mark IV (Thermo Fisher Scientific). All cryo-EM data were collected using a 200 kV Talos Arctica transmission electron microscope (Thermo Fisher Scientific) equipped with a K3 detector and BioQuantum energy filter (Gatan). All micrograph stacks were collected using EPU software in counting mode with a pixel size of 0.83 Å/pixel, a defocus range of −0.8 to −2.2 µm, and a dose rate 13.3 e$^-$/Å$^2$/s (total dose 40 e$^-$/Å$^2$, 50 frames).

### Cryo-EM data processing

The EM data processing workflow is presented in Supplementary Figs. 16–24. The three datasets with 5055, 6503, and 2794 micrographs were collected for the PG-bound, both CHS- and peptide-bound and ADP·BeF$_3$-bound conformations, respectively. A similar strategy was employed for cryo-EM data processing for all datasets. Briefly, beam-induced motion correction of the movie stacks and dose weighting was performed on the raw micrographs using MotionCor2[74]. The contrast transfer function (CTF) was estimated using CTFFIND4[75]. A template-free autopicking method based on a Laplacian-of-Gaussian filter was used to yield an initial set of particles from ~30 micrographs in RELION v3.1[76]. The particles were extracted using a box size of 260 × 260 pixels and subjected to reference-free 2D classification. Good 2D class averages were used as templates to automatically pick particles from all the micrographs. Extracted particles were then exported to cryoSPARC v3.1.0 for further data processing[77]. After multiple rounds of 2D classification, the selected particles were used to generate an Ab-initio model for heterogenous 3D refinement with C1 symmetry. The particles in good 3D classes were subjected to one or two rounds of homogeneous 3D refinement. The best class was selected and then imported back to RELION using the csparc2star.py module within UCSF pyem[78]. All maps were further improved by Bayesian polishing[76]. The polished particles were re-imported into cryoSPARC for another round of homogeneous 3D refinement, followed by iterative global and local CTF refinement. The final 3D reconstruction was performed using non-uniform refinement with C1 (PG-bound, both CHS- and peptide-bound) or C2 (ADP·BeF$_3$-bound) symmetry[79]. Overall resolutions were estimated based on a gold-standard Fourier Shell Correlation (FSC) cutoff of 0.143 between the two independently refined half-maps[80]. Local resolution was computed from the two half-maps in cryoSPARC.

### Cryo-EM model building and refinement

The previously reported crystal structure of human ABCB10 (PDB code 3ZDQ)[81] was used as an initial model by fitting its individual domains into the map as rigid bodies in UCSF Chimera[82]. The model was then refined using PHENIX real space refinement with secondary structure restraints, rotamer restraints, Ramachandran restraints, and non-crystallographic symmetry (NCS) restraints[83]. Models were built manually in Coot[84] and further improved by iterative rounds of model building in Coot and refinement in PHENIX. Regions with poor density were modeled as poly-alanine. The final refined structure was validated using MolProbity[85]. Cryo-EM data collection, processing, and refinement statistics are summarized in Table S2. All figures in the manuscript were generated using UCSF Chimera, Chimera X[86], and PyMOL (https://pymol.org/2/).

### Reporting summary

Further information on research design is available in the Nature Research Reporting Summary linked to this article.

## Data availability

The atomic coordinates of the three structures have been deposited in the Protein Data Bank with accession codes 7V5D (PG-bound), 7VFI (both CHS- and peptide-bound), and 7V5C (ADP·BeF$_3$-bound). The cryo-EM density maps have been deposited in the Electron Microscopy Data Bank with accession codes EMD-31723 (PG-bound), EMD-31955 (both CHS- and peptide-bound), and EMD-31722 (ADP·BeF$_3$-bound). Source data are provided with this paper.

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

## Acknowledgements

We are grateful to Dr. Alexey Amunts of Stockholm University for his help in data processing and Dr. Julian Gross for the critical reading of the manuscript. We also thank Mrs. Su Jeong Kim (POSTECH) and Dr. Jin-Seok Choi (KAIST Research Analysis Center) for assistance in grid screening and data acquisition. This research was supported by grants (M.S.J.) from the National Research Foundation (NRF) funded by the Ministry of Science, ICT, and Future Planning of Korea (NRF-2017M3A9F6029753, NRF-2019M3E5D6063908, and NRF-2021M3A9I4022846), and by a Ph.D. fellowship (J.G.P.) from the NRF funded by the Ministry of Science, ICT, and Future Planning of Korea (NRF-2022R1A6A3A13064599).

## Author contributions

J.G.P. and M.S.J. designed the experiments. S.J. performed gene cloning. J.G.P. and H.H. purified proteins. J.G.P., S.K., and J.W.K. determined the cryo-EM structures. J.G.P., E.J., and H.H. performed the biochemical studies. S.H.C. and D.S.M. prepared the figures and M.S.J. wrote the manuscript with the help of J.G.P.

## Competing interests

The authors declare no competing interests.
