## [Peer Review File · Nature Communications]

REVIEWER COMMENTS

Reviewer #1 (Remarks to the Author):

The manuscript of Park et al. describes the single particle cryo EM structures of mouse TAPL (ABCB6), a lysosomal ABC transporter specific for peptides ranging from 6 to more than 50 residues. To obtain the first structural information for TAPL, the authors purified mouse TAPL and deleted the N-terminal TMD0 domain. The protein was expressed in insect cells, solubilized in DM and purified in detergent solution (with or without cholesteryl hemi-succinate (CHS)) or reconstituted in nanodiscs (ND). The basal ATPase activity was higher in ND than in detergent (factor of 1.8 for v_{max} , while K_M remained similar).

Here the authors conclude (line 106 / 107) that the lipid environment is important for turnover. This might or might not be the case. It is well known the ATPase activity of ABC transporters in ND and detergent is different. If it is indeed the lipid environment, the authors have to include data that reports on the ATPase turnover in ND of different lipid composition.

Line 119: the authors make a valid point about ATPase activity and potential endogenous substrate(s). However, this issue can be addressed. If for example a lipid is the substrate and the substrate stimulates ATPase activity then ATPase activity over time is an important read-out. I like to explain this for CHS. If CHS is indeed a transport substrate, ATPase activity in the presence of CHS in a reconstituted system must change over time. As observed by the authors, CHS stimulates ATPase activity. Over time, the amount of CHS in the transport-competent state will be reduced due to the transport function of TAPL and stimulation should decrease over time to levels as if no CHS is around. To latter corresponds to the state when all CHS is transported.

The authors moved one step forward and employed a dithionite quenching assay to demonstrate transport of NBD-labeled lipids by liposome reconstituted TAPL. Transport was determined to be between 2.5 and 7.3 % depending on the lipid headgroup (line 134-135 and Figure 2). Per se this is convincing and would demonstrate for the first time that TAPL is not only a peptide transporter but also a lipid floppase. The fact that a preference for PE was observed is important as it demonstrates a difference between individual NBD-labeled lipids. However, in light of the history of NBD-labeled lipids in ABC transporter research (see for example ABCB1), the data does not doubtlessly demonstrate that the lipid moiety is the substrate and not the NBD moiety. I can only urge the authors to do assay with non-labeled lipids as used for example for MDR3 / ABCB4. Only then lipid transport can be demonstrated without doubt.

I also miss an ATPase deficient mutant of TAPL in these assays. It is known that already the presence of a single alpha helix can induce lipid flip / flop in liposomes. Thus, the systems have to be compared to a proper negative control.

Structures of the Δ TMD0 construct of TAPL were determined in the phosphatidyl-ethanolamine (PE), CHS / peptide (9mer) and the ADP BeF3 trapped state at overall resolutions ranging from 3.2 – 4.0 Å. Δ TMD0

reconstituted in ND and in the absence of peptide and / or lipids was determined at 3.4 Å overall resolution in the inward-facing (IF) state.

Line 157 / 159: are the authors saying that the central and upper cavities resemble the situation in ABCG2? If so please state. If not, please explain the difference between TAPL and ABCG2.

The overall structure resemble the well-known structure of ABC transporters belonging to the B subfamily, which is no big surprise and contains no real new information.

The peptide-bound state was determined for DM-solubilized Δ TMD0 in the presence of CHS at 4.0 Å overall resolution. Despite the low resolution, the authors successfully build the bound peptide explaining the binding preference for TAPL. The observed architecture was supported by mutational and transport studies of fluorescently labeled peptide by a liposome-based assay highlighting the importance of Y405. This residues interacts with both, the N- and C-termini of the bound peptide.

However, I miss the single Y405F mutant in Figure 4D. Is there a rational to not include this mutant in the analysis?

Finally, the authors found additional density in the upper cavity of Δ TMD0 reconstituted in ND. Based on the lipid composition of the ND, the authors build a PE molecule into the additional density. The interaction between lipid and Δ TMD0 is entirely hydrophobic. This is used by the authors to state that “This binding mode explains the preferential binding of neutral PE to Δ TMD0” (line 224- 225). Here, I disagree! PE is overall neutral, but zwitterionic in nature. The same is true for PC. But PC shows a lower transport rate in Figure 2. How is this possible? This point has to be clarified since the structural and functional data disagree at the moment! In other words, if ND would contain PC would PC bind as well indicating that TAPL is rather promiscuous for lipids? This is what the functional assay says.

Additionally, the density in Figure 5A is not of sufficient quality to allow the reviewer to judge the quality of the fit. The density in Figure S5B supports the notion that PE is bound, but how does the fit for PC or PG (the second component of the ND preparation)?

In a similar line of arguments, the authors identified three CHS molecules in Δ TMD0 (line 231) purified in DM micelles in the presences of CHS. CHS was verified by TLC. Again, I see a discrepancy between the structural and functional data. Sterols were used in Figure 2 as well but the transport rate was lower than for PE. If no three CHS molecules are transported in one step, but only one PE molecule, how can the functional data be explained?

The outward-facing (OF) state was obtained for Δ TMD0 reconstituted in ND in the presence of Mg^{2+} /ADPBeF3 and a 9mer peptide (line 245). An incubation with ADPBeF3 is in my opinion rather unusual as commonly one hydrolysis cycle is used to obtain the ADPBeF3 trapped state. Nevertheless, this incubation resulted in the OF structure at an overall resolution of 3.2 Å. The central cavity is now closed and the upper cavity widely open and no density for a peptide or lipid was detected.

Line 325: the statement that ATP binding induces the OF is obvious and properly right, but not supported by the data presented in this manuscript. No ATP bound state has been reported, only the ADP*BeF3 trapped stated. Therefore, it is not known what induces the OF state – ATP binding or hydrolysis (whether it is pre- or post-hydrolysis)!.The author should include this point in a revised version.

Line 285 – 287: I disagree that substrate binding induces the OF switch for members of the ABCB sub-family. It is substrate and ATP binding that induces the switch. Thus, TAPL does not break this generalization as the ATP-bound state has not been described.

In summary, this is an interesting manuscript that might be important for Nature Communications. However, the points raised above have to be clarified first before a final decision should be made.

Reviewer #2 (Remarks to the Author):

The authors report novel aspects of function and structure of TAPL as determined by flip-flop assay and cryo-EM study of TAPL(Δ TMP0) although the static data of cryo-EM does not necessarily show the transport. All these results seem to deserve publishing in Nature Communication. However, the manuscript is still in premature stage. Authors should add and modify following points to improve the report.

1) Line 23: The description “phosphoethanolamine (PE)-bound” is odd. In the text, authors describe “the bound PE” (Line 222), and they abbreviated phosphatidyl ethanolamine as PE (and also NBD-labelled PE) (Lines 218 and 546, respectively).

2) Line 397: Is the PBS buffer suitable for ATPase reaction, since probably high phosphate concentration in PBS inhibits ATP hydrolysis required for transport?

3) Lines 137-139: The finding is novel. However, it is not conclusive whether the phenomenon occurs coupled with ATP hydrolysis. Effects of ATPase inhibitor and/or AMP-PNP should be determined, which results would be helpful to discuss the cryo-EM data presented. Furthermore, it is also important to show whether the lipid floppase activity is inhibited in the presence of peptide or not.

4) Line 263-264: What stabilizes the hydrophobic residues in the outward-facing state when those residues move as shown in supplementary Fig.7b?

5) Authors should discuss structure relationship between TAPL and TAP(TAP1/TAP2). They showed in Fig.4C that Y405 of TAPL is conserved in TAP1. However, this residue is not in TAP2, although they do not indicate. F449 is not conserved in TAP1 and TAP2. Among G343, G459, G464 and G468, only G343 is

conserved in TAP1. As for Q298 and E456, Q298 is changed to Asp in TAP1, although E456 is conserved in TAP1 but changed to Gln in TAP2.

6) To figure legends, authors should indicate clearly whether the deltaTMD0 used are purified in the presence of CHS or not.

7) Several in the results should be clearly indicated (see followings).

Line 92-93: The Ref.23 could be deleted since it deals TAP1/2 complex but not TAPL. The “Refs. 5 and 25” and “Refs. 5, 13 and 24” may be cited separately in independent TMO holding and ATP hydrolysis and peptide transport by CoreTAPL, respectively.

Lines 96-102: Authors cited the references 2, 12, 13. However, only Ref. 2 uses an identical 9-mer peptide RRYQNSTC(fluorescein labelled)L. The 18-mer RRYQNSTELRRYQNSTC5-IAFL is related to the one in Ref.24. The 9-mer peptide RRYQKSTEL (Line 173) is used in Ref.2, 24. So please correct description appropriately.

Lines 110-111: Authors concluded that mouse deltaTMD0 and full-length human TAPL have similar biochemical characteristics. However, they did not compare the kinetic parameters for ATP hydrolysis and peptide transport (especially 9-mer). Modification of the description is needed.

Line 213: It is kind if Fig.3 and Supplementary Fig.4 are cited like “nanodisc-reconstituted DTMD0 structure in Fig.3 and supplementary Fig.4”.

As for Fig4d, it would be confirmative if authors showed that the Y405F/N301 mutant has normal ATPase activity.

Line 276: Is the description of “BeF3 of the Walker A loop” correct?

Supplementary Fig.2: The region of higher molecular size (more than 75 kDa) of top left panel of “b” (deltaTMD0 in nanodisc) should be shown similarly to other panels.

8) Materials and Methods: Biochemical and molecular biological parts should be described more clearly (see followings).

Lines 333-342: The followings should be indicated.

*Registered number of mouse TAPL cDNA sequence in NCBI (also human TAPL cDNA sequence in the legend to Supplementary Fig.1).

*Sequences of PCR primers used for amplification of TAPL cDNA and PCR mutagenesis.

*The method for site-directed mutagenesis (and also the mutagenesis kit if bought).

*Origins of cultured cells.

Lines 343-355:

*Suppliers (DNase I, PMSF, thrombin) should be mentioned, and also others: IPTG (Line 363), pyruvate kinase, lactate dehydrogenase, phosphoenolpyruvate (Lines 408-409), NBD-lipids (Line 421).

*HEPES -> HEPES-NaOH (also in other places)

*Was the solubilization of the membrane (Line 346) carried out using cell lysate directly after sonication without precipitation of the membranes?

*As for sonication (Line 346), type of sonifier and condition (output) should be indicated.

*As for washing buffer (Line 350), composition of the buffer is not clear.

Line 362: Ref is required for Luria-Bertani medium.

Lines 374-384: Reference for proteoliposome preparation, composition of 1xPBS (Line 377), and sonication condition (Line 378) together with a type of apparatus should be described.

Line 392: Show the type of HPLC column.

Lines 396-403:

*Please show the assay method for protein amount (Line 396), the volume of peptide transport assay, and the type of microplate reader (Line 403).

*If the appropriate reference for transport assay is available, cite it.

Reviewer #3 (Remarks to the Author):

In this study, functional and structural approaches are used to provide insight into the working mechanism of the mouse ABC transporter TAPL as a peptide translocase and lipid floppase. The authors provide first structural data of the transporter in a peptide-bound stage and remarkably report on a lipid floppase activity upon reconstitution into liposomes. The claims are certainly novel and potentially impactful. However, there are major flaws in the current version of the manuscript.

Major issues:

- Many ATP-dependent transporters need to be re-lipidated to get back an active protein. Thus, lipid-dependent stimulation does not allow to conclude on the substrate. The author over interpret their observations. Furthermore, to claim sterol activation the authors would need to prepare nanodiscs based on phospholipids with and without sterol.
- The NBD-lipid transport assay is based on the principle that the dianion dithionite is membrane impermeant. Thus, dithionite only quenches the fluorescent probes in the outer leaflet of liposomes. Thus, upon addition of dithionite the fluorescence should reach a stable plateau. This condition is not fulfilled here and additional controls are required. First, mock reconstitutions for each NBD-lipids should be added to ensure that differences between the two conditions originate from the presence of protein. Upon adding dithionite, it is expected to observe ~ 50% reduction in fluorescence. Second, the presence of ATP can affect the lipid distribution in liposomes. Thus, proteoliposomes with lipid flippases/floppases need to be tested after incubation with Mg-ATP relative to control vesicles incubated with Na-ATP. Third, reconstitution and transport studies need to be repeated with a catalytically inactive mutants. Without these controls, it is not possible to assign the observed difference to active transport.
- The dithionite accessibility is quite similar for all NBD-probes tested. This is unusable ad NBD-cholesterol has been reported to undergo rapid transbilayer movement in liposomes. The author do not address this point in comparison to published data.
- The simplest explanation for the transport data represented is the leakage of the dianion dithionite through the vesicle membrane. Thus, tests for leakiness of the proteoliposomes are required (e.g. NBD-glucose entrapment followed by dithionite quenching of ABD-liposomes – as presented in Goren et al., 2014).

- Dialysis was used for reconstitution of the floppase-active proteoliposomes. This procedure seems unlikely to be sufficient for removal of DM, which has a very low CMC similar to DDM. Did the authors check for the complete removal of DM detergent in the proteoliposome sample?

- The structure of TAPL is resolved at 3.4 Å with the binding site at 4.0 Å, which is quite low and raises concerns about possible overinterpretation. In particular the lipid binding site requires further validation. For example, have the mutants (Y405F/N301A) be tested for lipid transport activity? Without such validation, the result is open to a possibility that the reported conformation is an artifact.

Further issues:

- While the authors show the protein content of the nanodiscs, the phospholipid composition is not reported. Have all E.coli lipids reconstituted according to the starting lipid composition?

- The methods section lacks several details: Specify amount of bio-beads and time for incubation during reconstitution, specify how much proteoliposome were used for floppase assay.

- A Coomassie-stained SDS-PAGE after reconstitution would also be necessary not only to check that protein is reconstituted but to check that the differences in the percentages of lipid transported are not due to differences in amount of protein reconstituted.

- In figure 2, a legend should be added for every graph since the colors change for every lipid. Otherwise, a discontinuous vs a continuous line should be added.

Comments from Committee (Bold) and Author Responses

Reviewer #1 (Remarks to the Author):

The manuscript of Park et al. describes the single particle cryo EM structures of mouse TAPL (ABCB9), a lysosomal ABC transporter specific for peptides ranging from 6 to more than 50 residues. To obtain the first structural information for TAPL, the authors purified mouse TAPL and deleted the N-terminal TMD0 domain. The protein was expressed in insect cells, solubilized in DM and purified in detergent solution (with or without cholesteryl hemi-succinate (CHS)) or reconstituted in nanodiscs (ND). The basal ATPase activity was higher in ND than in detergent (factor of 1.8 for V_{max} , while K_M remained similar).

1. Here the authors conclude (line 106 / 107) that the lipid environment is important for turnover. This might or might not be the case. It is well known the ATPase activity of ABC transporters in ND and detergent is different. If it is indeed the lipid environment, the authors have to include data that reports on the ATPase turnover in ND of different lipid composition.

>> As suggested, we have measured the lipid composition-dependent ATPase kinetics of Δ TMD0. The data are presented in Figure 1c and discussed on pages 5-6.

2. Line 119: the authors make a valid point about ATPase activity and potential endogenous substrate(s). However, this issue can be addressed. If for example a lipid is the substrate and the substrate stimulates ATPase activity then ATPase activity over time is an important read-out. I like to explain this for CHS. If CHS is indeed a transport substrate, ATPase activity in the presence of CHS in a reconstituted system must change over time. As observed by the authors, CHS stimulates ATPase activity. Over time, the amount of CHS in the transport-competent state will be reduced due to the transport function of TAPL and stimulation should decrease over time to levels as if no CHS is around. To latter corresponds to the state when all CHS is transported.

>> We appreciate the helpful comments and suggestions. However, in our view, the proposed experiment is unlikely to be useful because the amounts of phospholipid in the liposome membranes are much greater than the amount of CHS.

>> Instead, we investigated whether the addition of CHS (or phospholipids) to detergent-purified Δ TMD0 stimulated its ATPase activity in a dose-dependent manner. The results suggest that lipids bind to TAPL and that their transport is coupled to ATP hydrolysis. We discuss these data in the revised text (page 7), and have added Figures 2a-b.

3. The authors moved one step forward and employed a dithionite quenching assay to demonstrate transport of NBD-labeled lipids by liposome reconstituted TAPL. Transport was determined to be between 2.5 and 7.3 % depending on the lipid headgroup (line 134 7 135 and Figure 2). Per se this is convincing and would demonstrate for the first time that TAPL is not only a peptide transporter but also a lipid

floppase. The fact that a preference for PE was observed is important as it demonstrates a difference between individual NDB-labeled lipids. However, in light of the history of NBD-labeled lipids in ABC transporter research (see for example ABCB1), the data does not doubtlessly demonstrate that the lipid moiety is the substrate and not the NBD moiety. I can only urge the authors to do assay with non-labeled lipids as used for example for MDR3/ABCB4. Only than lipid transport can be demonstrated without doubt (Fig. 2c).

>> As mentioned above (# point 2), we have confirmed that non-labeled lipid itself was sufficient to stimulate the ATPase activity of Δ TMD0. In addition, its NBD-derivative displayed a similar level of stimulation, indicating that NBD conjugation does not impair the intrinsic lipid transport activity of Δ TMD0. The results are presented in Figures 2a-b and Supplementary Figure 4b in the revised manuscript.

4. I also miss an ATPase deficient mutant of TAPL in these assays. It is known that already the presence of a single alpha helix can induce lipid flip / flop in liposomes. Thus, the systems have to be compared to a proper negative control.

>> We agree with this point. We now include a flip-flop analysis of the ATPase-deficient E664Q mutant as a negative control. As expected, unlike the wild-type, the E664Q mutant exhibited a defect in lipid flipping, indicating that TAPL-mediated lipid transport is driven by ATP hydrolysis. We include these data in Figures 2c-e.

5. Structures of the Δ TMD0 construct of TAPL were determined in the phosphatidylethanolamine (PE), CHS / peptide (9mer) and the ADP BeF3 trapped state at overall resolutions ranging from 3.2 – 4.0 Å. Δ TMD0 reconstituted in ND and in the absence of peptide and / or lipids was determined at 3.4 Å overall resolution in the inward-facing (IF) state.

Line 157 / 159: are the authors saying that the central and upper cavities resemble the situation in ABCG2? If so please state. Revise the text.

>> We have revised the text and added associated references. Please see page 9.

6. The overall structure resembles the well-known structure of ABC transporters belonging to the B subfamily, which is no big surprise and contains no real new information. The peptide-bound state was determined for DM-solubilized Δ TMD0 in the presence of CHS at 4.0 Å overall resolution. Despite the low resolution, the authors successfully build the bound peptide explaining the binding preference for TAPL. The observed architecture was supported by mutational and transport studies of fluorescently labeled peptide by a liposome-based assay highlighting the importance of Y405. This residue interacts with both, the N- and C-termini of the bound peptide. However, I miss the single Y405F mutant in Figure 4D. Is there a rationale to not include this mutant in the analysis?

>> In the revised version, we use the Y405A mutant as a negative control in the peptide

transport assay, and confirm that, compared to the wild-type, the Y405A mutant shows a >73-fold decrease in peptide affinity and complete loss of peptide transport activity. The results are presented in Figures 4d-5, and discussed in the text, page 11.

7. Finally, the authors found additional density in the upper cavity of Δ TMD0 reconstituted in ND. Based on the lipid composition of the ND, the authors build a PE molecule into the additional density. The interaction between lipid and Δ TMD0 is entirely hydrophobic. This is used by the authors to state that “This binding mode explains the preferential binding of neutral PE to Δ TMD0” (line 224- 225). Here, I disagree! PE is overall neutral, but zwitterionic in nature. The same is true for PC. But PC shows a lower transport rate in Figure 2. How is this possible? This point has to be clarified since the structural and functional data disagree at the moment! In other words, if ND would contain PC would PC bind as well indicating that TAPL is rather promiscuous for lipids? This is what the functional assay says.

>> We agree with this point. In this revision, we have found that various phospholipids and CHS act to stimulate the ATPase activity of Δ TMD0 in a concentration-dependent manner. These account for the promiscuous recognition of Δ TMD0 for structurally diverse lipids. We discuss the results on page 12, and present them in Figures 2a-b.

8. Additionally, the density in Figure 5A is not of sufficient quality to allow the reviewer to judge the quality of the fit. The density in Figure S5B supports the notion that PE is bound, but how does the fit for PC or PG (the second component of the ND preparation)? (Also build PC or PG, Supplementary Fig. 5).

>> We have added atomic models of PC, PG and PS as Figure 6 and Supplementary Figures 8-9, as suggested.

9. In a similar line of arguments, the authors identified three CHS molecules in Δ TMD0 (line 231) purified in DM micelles in the presences of CHS. CHS was verified by TLC. Again, I see a discrepancy between the structural and functional data. Sterols were used in Figure 2 as well but the transport rate was lower than for PE. If no three CHS molecules are transported in one step, but only one PE molecule, how can the functional data be explained?

>> Unfortunately we cannot offer an explanation as, despite extensive efforts, we have not been able to accurately measure Δ TMD0-mediated cholesterol transport activity, probably due to its rapid spontaneous movement across the liposome bilayer.

10. The outward-facing (OF) state was obtained for Δ TMD0 reconstituted in ND in the presence of Mg^{2+} /ADPBeF3 and a 9mer peptide (line 245). An incubation with ADPBeF3 is in my opinion rather unusual as commonly one hydrolysis cycle is used to obtain the ADPBeF3 trapped state. Nevertheless, this incubation resulted in the OF structure at

an overall resolution of 3.2 Å. The central cavity is now closed and the upper cavity widely open and no density for a peptide or lipid was detected.

Line 325: the statement that ATP binding induces the OF is obvious and properly right, but not supported by the data presented in this manuscript. No ATP bound state has been reported, only the ADP*BeF3 trapped state. Therefore, it is not known what induces the OF state – ATP binding or hydrolysis (whether it is pre- or post-hydrolysis)!. The author should include this point in a revised version (Revise the text).

>> We have revised the conclusion as suggested. Please see pages 17-18.

11. Line 285 – 287: I disagree that substrate binding induces the OF switch for members of the ABCB sub-family. It is substrate and ATP binding that induces the switch. Thus, TAPL does not break this generalization as the ATP-bound state has not been described.

>> We have re-written the text to clarify this statement. Please see page 16.

=====
Reviewer #2 (Remarks to the Author):

The authors report novel aspects of function and structure of TAPL as determined by flip-flop assay and cryo-EM study of TAPL(Δ TMP0) although the static data of cryo-EM does not necessarily show the transport. All these results seem to deserve publishing in Nature Communication. However, the manuscript is still in premature stage. Authors should add and modify following points to improve the report.

1. Line 23: The description “phosphoethanolamine (PE)-bound” is odd. In the text, authors describe “the bound PE” (Line 222), and they abbreviated phosphatidyl ethanolamine as PE (and also NBD-labelled PE) (Lines 218 and 546, respectively).

>> We have revised the text.

2. Line 397: Is the PBS buffer suitable for ATPase reaction, since probably high phosphate concentration in PBS inhibits ATP hydrolysis required for transport?

>> We fully understand the concern. However, many workers have measured the peptide transport activities of TAPL using liposomes resuspended in PBS buffer (JBC, 2008, 283, 17083-91). Moreover, when we repeated these assays in HEPES buffer (pH 8), Δ TMD0 showed similar levels of peptide transport activity (data not shown). This suggests that the high phosphate concentration of PBS does not affect Δ TMD function.

3. Lines 137-139: The finding is novel. However, it is not conclusive whether the phenomenon occurs coupled with ATP hydrolysis. Effects of ATPase inhibitor and/or AMP-PNP should be determined, which results would be helpful to discuss the cryo-EM data presented. Furthermore, it is also important to show whether the lipid floppase activity is inhibited in the presence of peptide or not.

>> As suggested, we have examined the lipid flip-flop activity of Δ TMD0 under non-ATP hydrolytic conditions (Na^+/ATP) and in the presence of 9-mer peptide. The data indicate that Δ TMD0-mediated lipid transport activity is significantly reduced under these conditions. We discuss these results on page 8, and present them in Figures 2c-e.

4. Line 263-264: What stabilizes the hydrophobic residues in the outward-facing state when those residues move as shown in supplementary Fig.7b?

>> We discuss this point on pages 15.

5. Authors should discuss structure relationship between TAPL and TAP (TAP1/TAP2). They showed in Fig.4C that Y405 of TAPL is conserved in TAP1. However, this residue is not in TAP2, although they do not indicate. F449 is not conserved in TAP1 and TAP2. Among G343, G459, G464 and G468, only G343 is conserved in TAP1. As for Q298 and E456, Q298 is changed to Asp in TAP1, although E456 is conserved in TAP1 but changed to Gln in TAP2.

>> We now include a section of the revised manuscript discussing the structural relationship between TAPL and the TAP1/2 complex, as suggested; see page 14 and Supplementary Figure 11.

6. To figure legends, authors should indicate clearly whether the deltaTMD0 used are purified in the presence of CHS or not.

>> We have revised the figure legends to clarify this point.

7. Several in the results should be clearly indicated (see followings).

(1) Line 92-93: The Ref.23 could be deleted since it deals TAP1/2 complex but not TAPL. The "Refs. 5 and 25" and "Refs. 5, 13 and 24" may be cited separately in independent TM0 holding and ATP hydrolysis and peptide transport by CoreTAPL, respectively.

>> We have corrected the references.

(2) Lines 96-102: Authors cited the references 2, 12, 13. However, only Ref. 2 uses an identical 9-mer peptide RRYQNSTC(fluorescein labelled)L. The 18-mer RRYQNSTELR RYQNSTC(5-IAF)L is related to the one in Ref.24. The 9-mer peptide RRYQKSTEL (Line 173) is used in Ref.2, 24. So please correct description appropriately.

>> We have made the suggested changes.

(3) Lines 110-111: Authors concluded that mouse deltaTMD0 and full-length human TAPL have similar biochemical characteristics. However, they did not compare the kinetic parameters for ATP hydrolysis and peptide transport (especially 9-mer).

Modification of the description is needed.

>> We agree with this comment and have revised the text (page 6).

(4) Line 213: It is kind if Fig.3 and Supplementary Fig.4 are cited like “nanodisc-reconstituted DTMD0 structure in Fig.3 and supplementary Fig.4”.

>> We have revised the text, as suggested.

(5) As for Fig4d, it would be confirmative if authors showed that the Y405F/N301 mutant has normal ATPase activity.

>> We have dealt with this matter above. Please see our response to Reviewer 1, point 6.

(6) Line 276: Is the description of “BeF3 of the Walker A loop” correct?

>> We have corrected this point.

(7) Supplementary Fig.2: The region of higher molecular size (more than 75 kDa) of top left panel of “b” (deltaTMD0 in nanodisc) should be shown similarly to other panels.

>>We have made the change.

8. Materials and Methods: Biochemical and molecular biological parts should be described more clearly (see followings).

(1) Lines 333-342: The followings should be indicated.

***Registered number of mouse TAPL cDNA sequence in NCBI (also human TAPL cDNA sequence in the legend to Supplementary Fig.1).**

***Sequences of PCR primers used for amplification of TAPL cDNA and PCR mutagenesis.**

***The method for site-directed mutagenesis (and also the mutagenesis kit if bought).**

***Origins of cultured cells.**

>> We have modified the Methods section as suggested, and now include Supplementary Table 1 showing the primers used in this study.

>> The method of site-directed mutagenesis was already mentioned on page 18 (i.e. “Mutations were introduced by polymerase chain reaction (PCR)-based site-directed mutagenesis”).

(2) Lines 343-355:

***Suppliers (DNase I, PMSF, thrombin) should be mentioned, and also others: IPTG (Line 363), pyruvate kinase, lactate dehydrogenase, phosphoenolpyruvate (Lines 408-409), NBD-lipids (Line 421).**

***HEPES -> HEPES-NaOH (also in other places)**

***Was the solubilization of the membrane (Line 346) carried out using cell lysate directly after sonication without precipitation of the membranes?**

***As for sonication (Line 346), type of sonifier and condition (output) should be indicated.**

***As for washing buffer (Line 350), composition of the buffer is not clear.**

>> We now provide the relevant details.

(3) Line 362: Ref is required for Luria-Bertani medium.

>> We have added the associated reference.

(4) Lines 374-384: Reference for proteoliposome preparation, composition of 1xPBS (Line 377), and sonication condition (Line 378) together with a type of apparatus should be described.

>> We have revised the description in Methods, as suggested.

(5) Line 392: Show the type of HPLC column.

>> We have revised the text.

(6) Lines 396-403:

***Please show the assay method for protein amount (Line 396), the volume of peptide transport assay, and the type of microplate reader (Line 403).**

***If the appropriate reference for transport assay is available, cite it.**

>> We now provide these details (Page 22), and have added an associated reference.

=====

Reviewer #3 (Remarks to the Author):

In this study, functional and structural approaches are used to provide insight into the working mechanism of the mouse ABC transporter TAPL as a peptide translocase and lipid floppase. The authors provide first structural data of the transporter in a peptide-bound stage and remarkably report on a lipid floppase activity upon reconstitution into liposomes. The claims are certainly novel and potentially impactful. However, there are major flaws in the current version of the manuscript.

Major issues:

1. Many ATP-dependent transporters need to be re-lipidated to get back an active protein. Thus, lipid-dependent stimulation does not allow to conclude on the substrate. The author overinterpret their observations. Furthermore, to claim sterol activation the authors would need to prepare nanodiscs based on phospholipids with and without

sterol.

>> We have dealt with this matter above. Please see Reviewer 1, # point 2.

2. The NBD-lipid transport assay is based on the principle that the dianion dithionite is membrane impermeant. Thus, dithionite only quenches the fluorescent probes in the outer leaflet of liposomes. Thus, upon addition of dithionite the fluorescence should reach a stable plateau. This condition is not fulfilled here and additional controls are required. First, mock reconstitutions for each NBD-lipids should be added to ensure that differences between the two conditions originate from the presence of protein. Upon adding dithionite, it is expected to observe ~ 50% reduction in fluorescence. Second, the presence of ATP can affect the lipid distribution in liposomes. Thus, proteoliposomes with lipid flippases/floppases need to be tested after incubation with Mg-ATP relative to control vesicles incubated with Na-ATP. Third, reconstitution and transport studies need to be repeated with catalytically inactive mutants. Without these controls, it is not possible to assign the observed difference to active transport.

>> We appreciate these helpful comments and suggestions. During the revision we repeated the NBD-lipid transport assays several times using liposomes composed of *E.coli* polar lipid extract, but it was difficult to obtain a stable baseline after addition of dithionite. As previously reported, it appears that the *E.coli* lipid membrane is more permeable to dithionite due to its high fluidity (Biochem J. 2010, 429, 195–203).

>> After screening liposomes of various lipid compositions, we found that liposomes composed of a 1:1 (mol/mol) mixture of *E. coli* lipids and egg PC showed no leakage even after two dithionite treatments (Supplementary Figure 5b). Using these non-leaky liposomes, we repeated the NBD-lipid flipping assays, and re-analyzed the relevant data.

>> We now include assays in the presence of Na⁺/ATP and with the ATPase-deficient E664Q mutant. As expected, there was no lipid transport activity, indicating that TAPL functions as an ATP-driven lipid floppase. We discuss these results in the text (please see page 7-8), and present them in Figures 2c-e.

3. The dithionite accessibility is quite similar for all NBD-probes tested. This is unusable and NBD-cholesterol has been reported to undergo rapid transbilayer movement in liposomes. The author does not address this point in comparison to published data.

>> This matter is no longer an issue, since in the revised text we only use NBD-PS to examine the role of Δ TMD0 as a floppase.

>> We have added a discussion of the rapid transbilayer movement of NBD-cholesterol on page 8.

4. The simplest explanation for the transport data represented is the leakage of the dianion dithionite through the vesicle membrane. Thus, tests for leakiness of the proteoliposomes are required (e.g. NBD-glucose entrapment followed by dithionite quenching of ABD-liposomes – as presented in Goren et al., 2014).

>> We agree that this is an important question. To test for leakage of the liposomes, dithionite was re-added to the liposomes once a stable baseline had formed after its first addition. We hypothesized that if the liposomes were permeable to dithionite, there would be another rapid decrease in fluorescence. Of the various liposomes tested, liposomes composed of a 1:1 (mol/mol) mixture of *E. coli* lipid and egg PC showed a stable baseline after the second addition of dithionite, whereas the other liposomes did not. These data are presented in Supplementary Figure 5c.

5. Dialysis was used for reconstitution of the floppase-active proteoliposomes. This procedure seems unlikely to be sufficient for removal of DM, which has a very low CMC similar to DDM. Did the authors check for the complete removal of DM detergent in the proteoliposome sample?

>> In the revised version, we test how much bio-beads is needed to completely remove the detergents, since the bio-beads protocol generated non-leaking liposomes. The results are presented in Supplementary Figure 5a.

6. The structure of TAPL is resolved at 3.4 Å with the binding site at 4.0 Å, which is quite low and raises concerns about possible overinterpretation. In particular, the lipid binding site requires further validation. For example, have the mutants (Y405F/N301A) be tested for lipid transport activity? Without such validation, the result is open to a possibility that the reported conformation is an artifact.

>> In the revised version, we use the Y405A mutant, instead of Y405F/N301A, as a negative control for the functional assays. We confirmed that the Y405A mutant with reduced ATPase activity exhibited no substrate (peptide or lipid) transport activity and had an 73-fold reduction in peptide affinity compared to the wild-type. The results are shown in Figures 4d-5 and Supplementary Figure 7, and discussed in the text, page 11.

Further issues:

7. While the authors show the protein content of the nanodiscs, the phospholipid composition is not reported. Have all *E.coli* lipids reconstituted according to the starting lipid composition?

>> We have revised the Methods description to clarify this point.

8. The methods section lacks several details: Specify amount of bio-beads and time for incubation during reconstitution, specify how much proteoliposome were used for floppase assay.

>> We have clarified these points.

9. A Coomassie-stained SDS-PAGE after reconstitution would also be necessary not only to check that protein is reconstituted but to check that the differences in the

percentages of lipid transported are not due to differences in amount of protein reconstituted.

>> We have added SDS-PAGE gels showing the amount of protein reconstituted into the nanodiscs and liposomes. Please refer to Supplementary Figures 3 and 5d, respectively.

10. In figure 2, a legend should be added for every graph since the colors change for every lipid. Otherwise, a discontinuous vs a continuous line should be added.

>> We have made the changes as suggested.

REVIEWER COMMENTS

Reviewer #1 (Remarks to the Author):

All points raised have been adequately addressed by the authors. I recommend acceptance of the revised version of the manuscript.

Reviewer #2 (Remarks to the Author):

Most of the comments to the original manuscript are considered by the authors. However, the revised manuscript in the first half has completely remodeled since contradict results of floppase activities to the original manuscript are replaced in the revised one without any explanation to the reviewers. Furthermore, there are still many ambiguities in the revised manuscript. I would like to suggest that the description of the revised manuscript must be carefully checked including the figure titles and legends (also supplementary figures), make confirmation experiments and logically re-written the manuscript.

1. All the phospholipids used in the study should be carefully checked by means of two-dimensional TLC. If impurities and lyso-phospholipids are contaminated, side-effects may occur: lyso-phospholipids may affect the dispersion of micelle form, and act as detergent in membranous structure. Especially, transport assay could be prohibited by the detergent action of lyso-phospholipids. (Reviewer experienced that the labels had been stucked differently for vials of synthetic phospholipids from a famous lipid supplier, and so-called purified PS from another famous was found to be crude extract.)

1) Authors did not answer the Comment 8 (2) as to NBD-lipids. Please describe the supplier of NBD-lipids or person donated, and their purity. How NBD is conjugated? It is described that NBD-lipids have a variety of acyl chain-labeled (Lines 147-148). However, it is not certain whether the labeling pattern is similar in different phospholipids or not. Please indicate these points in detail.

2) Although it may be true that the NBD-conjugation did not affect significantly on the ATPase activities of DM-purified Δ TMD0 (supplementary Fig.4b), it is too speculative if authors say that NBD-conjugation did not influence the functional properties of lipids as substrates of TAPL (Lines 148-149). Only floppase activity for NBD-PS was demonstrated in the revised manuscript.

2. In contrast to original manuscript which showed that most lipids (PE, PS, PC, PG, sphingomyelin and cholesterol) flopped, only PS is demonstrated to move in the revised manuscript. The logic of identification of PG in the cryo-EM is exclusion of the possibility of PE and CL (Lines 248-250). However, it is also possible that phospholipid had been bound before solubilization of Δ TMD0.

Authors did not discuss the phospholipid and sterol contents in the lysosomal membrane. Biological meaning of phospholipid floppase activity of TAPL in lysosome should be discussed briefly although authors also showed the possibility that other phospholipids such as PS and PC could bind to lipid binding site (Lines 257-260). Useful information would be found in FEBS J. (2021) 288, 4168 and Int. J. Mol. Sci. (2019) 20, 2167.

3. Since the study is not only for the researchers of protein structure analyses but also many other fields such as biochemistry and medical sciences and so on, authors should mention their study carefully and politely (Comment 8) to those who are able to understand the story if this paper would become acceptable.

1) As for Fig.1, the Fig.1a is clear since transport assay is carried out using proteoliposome. The Fig1b is probably the results of DM-purified micelles from the legend (Line 644). Both DM micelles and nanodiscs are used in Fig1c as indicated in the legend (Line 646). However, how nanodiscs are prepared with PO (1-palmitoyl-2-oleoyl)-phospholipids (Fig.1c) were not indicated in the Materials and Methods (Lines 427 – 443). The supplier and purity of PO-phospholipids were not mentioned. Please add these points.

2) As for Fig.2(a, b), detergent-purified (Line 132) and DM-purified (Lines 648, 655) Δ TMD0 are indicated. Since the title of Fig.2 is “transport activity” (Line 648), I would like to suggest that Fig.2(a, b) should be moved to Fig.1 (as d, e, respectively) or modify the figure title.

Is the “supplementary Fig.3” (Line 134) correct? Probably, supplementary Fig.4b is correct. Since supplementary Fig4a is schematic of proteoliposome and title is flip-flop, appear of supplementary Fig4b in the same figure is odd. The title of supplementary Fig.4 should be modified.

Since supplementary Fig3 (probably Fig4b) is appeared together with Fig. 2a,b (Line134), the three small figures (Fig. 2a, b and supplementary Fig. 4b) may be the results of DM-purified Δ TMD0. However, this should be clearly indicated in the legend to supplementary Fig.4. Otherwise, the readers would misunderstand that the samples of supplementary Fig.4b were proteoliposomes like supplementary Fig. 4a. Supplementary Fig.3 should be cited in an appropriate position (probably Lines 107-121 or 427-443).

3) To Comment 6, not responded well: in above 1) and 2), DM-purified Δ TMD0 seems to be solubilized and purified without CHS. Please indicate whether the CHS is present or not during solubilization and purification in each sample used. It must be strengthened that the hysteresis strongly effects on the mode of interaction between membrane proteins and lipids.

What kind of sample of Δ TMD0 (solubilized and purified in the presence CHS?) was used to prepare nanodisc in the presence of CHS?

As for proteoliposome preparation, how was the purified protein (Lines 454-456, 469) prepared (solubilization and purification without CHS)?

As for cryo-EM sample preparation, authors indicated DM/CHS-purified Δ TMD0 (Line 567) and also Δ TMD0 purified in DM/CHS micelles (Line 201, 261). Probably Δ TMD0 was solubilized and purified in the presence of CHS. If so, authors should define Δ TMD0 as DM-purified Δ TMD0 (solubilized and purified in the absence of CHS) or DM/CHS-purified Δ TMD0 (or DM/CHS) (solubilized and purified in the presence of CHS) (Lines 416-422).

Meaning of the detergent-purified Δ TMD0 is not clear (Line 543). Does it mean DM-purified Δ TMD0 or DM/CHS-purified Δ TMD0?

4) The Δ TMD0 in nanodisc without peptides and nucleotides was first subjected to cryo-EM analysis (Fig. 3 and Supplementary Fig. 6a-c). Then, authors analyzed DM/CHS micelles in the presence of 9-mer peptide. Further, they compared peptide bound structure with that without peptide in Supplementary Fig. 6d. In the legend to Supplementary Fig. 6d, it should be mentioned that cryo-EM structures of PG-bound nanodisc and CHS & peptide-bound DM/CHS micelles are compared.

5) As for Supplementary Fig. 7a (Line 235), condition of samples for ATPase measurement is difficult to understand. Do authors use proteoliposomes, NBD-PS embedded-proteoliposomes, DM-micelles or nanodiscs? How do they add POPG and CHS?

4. Authors evaluated the leakiness of proteoliposomes as indicated in supplementary Fig. 5c. However, purpose and results of experiments of supplementary Fig. 5a-c were not explained. Please mention these essences in appropriate site (e.g. Lines 513-531).

Authors indicated the method of how to prepare NBD dye-labeled proteoliposomes in the original manuscript. However, in the revised manuscript this portion is omitted. Since authors made efforts to prepare the tightly sealed proteoliposomes for the assay using dithionite, why they include the important improved method in Method section. It is also necessary to add what kinds of NBD-lipid is added in the legend to supplementary Fig 5c. In Line 169, (Figs.2c, d) should be inserted.

5. Lines 375-376: "the TAP complex has similar preferences to TAPL in terms of peptide length" is not correct. As the authors describe in Introduction (Lines 54 – 55), TAP complex favors peptides of 8-12 residues, whereas TAPL transports a broad spectrum of peptides, ranging from 6 to 59 residues. Although three reviewers did not indicate this part in the original manuscript, revision is necessary in the final version. If difference in central region of peptide binding sites of TAP and TAPL could be imagined, please discuss.

6. Lines 233-234: Briefly explain that the affinity of peptide binding to Δ TMD0 (wild and Y405) is determined by means of microscale thermophoresis.

7. Lines 302-303: G452 of TAPL is also conserved in TAP2.

8. As for peptide sequences (Comments 7 (2)), Please add also Ref.2 [JBC (2005) 280, 23631.] since peptide sequences [RRYQKSTEL and RRYQNSTC(Φ)L] were shown in the Ref.2 (Lines 98, 202).

Reviewer #3 (Remarks to the Author):

The authors have resubmitted a substantially revised version of their previous manuscript with a number of new experiments added: lipid composition-dependent ATPase kinetics of Δ TMD0, lipid flip-flop activity of Δ TMD0 under non-ATP hydrolytic conditions (Na⁺/ATP), and flip-flop analysis of the ATPase-deficient E664Q mutant as a negative control. While the efforts of the authors are highly appreciated, there are still misleading statements and overinterpretations of the new results that should be addressed in a second revision as outline below.

(1) The authors measured now ATPase activity on both detergent-solubilized and nanodiscs-reconstituted TAPL. Essentially, all lipids tested stimulated the basal activity in both set-ups (PE could not be tested for technical reasons). Thus, the general lipid-dependent stimulation observed by the authors does not allow one to conclude on the substrate of TAPL. The author still over interpret their observations and should carefully adjust the text and discussion of the manuscript. Here are only few examples of misleading statements throughout the revised manuscript:

a. Lines 74-76: The sentence reading “Moreover, to the best of our knowledge, this is the first report that TAPL binds directly to endogenous lipids and flips them from the cytoplasmic leaflet of the lysosome to the luminal leaflet in an ATP-dependent manner” states a function in living cells which has not been reported in the scope of this manuscript. Instead, we suggest replacing this sentence into something in the following lines: “Moreover, to the best of our knowledge, this is the first report that

TAPL binds directly to endogenous lipids and transports a fluorescent lipid analogue in an ATP-dependent manner”.

b. The authors suggest a link between lipid stimulation and lipid transport, and state “The results suggest that lipids are TAPL substrates and their translocation is coupled to ATP hydrolysis (lines 140-141). The current data do not show this; all lipids stimulate ATPase activity but only NBD-PS is transported (without specific higher stimulation of ATPase activity). Along the lines, the title of Figure 2 is not precise: “Fig. 2. ATP-dependent lipid transport activity”; only in panel d, lipid transport is presented.

c. Statements such as “NBD conjugation does not impair the intrinsic lipid transport activity of Δ TMD0” are incorrect. The authors have only shown that NBD conjugation does not impair ATPase stimulation!

d. Lines 156-157: Based on the data from figure 2e, a more accurate interpretation would be to say that percentage of NBD-PS transport driven by a catalytically active Δ TMD0 was ~2%. This is based on that both in presence of the peptide or the catalytically inactive version also show a difference.

(2) Line 151-151: “In the presence of Mg^{2+} /ATP, the addition of 10 mM dithionite to Δ TMD-loaded proteoliposomes led to a fast drop in the fluorescence of NBD-PS to ~36% of the original value (Fig. 2c and Supplementary Figs. 5a, b)”

Supplementary Figs. 5a, b do not show this information given in the sentence. In fact, Supplementary Figs. 5a, b are not at all presented and discussed in the main text.

(3) In supplementary Figure 5, panel c the authors use liposomes, not proteoliposomes. As such, the authors only probe the permeability of protein-free membranes; this should be clearly indicated in the figure, legend and text. In the current main text of the revised version, this panel is not presented and explain.

(4) I cannot follow the argumentation of the authors on the different dithionite plateaus for liposomes and proteoliposomes. Why do not both vesicles have a 50/50% distribution of NBD-PS in the absence of ATP?

Furthermore, Which NBD-lipid has been used, C6 or C12? Please give the full chemical name of the lipid probe.

(5) Lipid transport activity was now tested along with crucial controls (ATPase-dead mutant, Na/Mg-ATP), showing that NBD-PS, a fluorescent analog of PS is transported. Whether TAPL transports endogenous lipids remains open. This should be clearly stated in the abstract and text of the manuscript. Strictly speaking, already the title overstates the data presented.

(6) From supplementary Figure 4 it is unclear if the ATPase activity of Δ TMD0 have been measured on vesicle-reconstituted protein. Please indicate this clearly and consider revising the figure title.

(7) Equal important, the authors should clear indicate the statistics to each panel in the main manuscript and in the supplement including the number of independent experiments performed, i.e. independent reconstitutions. For example, the authors state that the data indicate that Δ TMD0-mediated lipid transport activity is significantly reduced under these conditions (Figures 2c-e.) but have not included the statistical analysis and significance, number of measurements, and number of independent reconstitutions.

(8) I could not find careful discussion on lipids/lipid translocation/floppase in relation of the structure and functional findings of Δ TMD0 in the Discussion section.

(9) Regarding the protein reconstitution, essential information is still missing. In Supplementary Figures 3 and 5, the authors should determine quantities of the protein based on the BSA standard. What was the final lipid to protein ratio that was achieved for the proteoliposomes. Such information is crucial for reproducing the experiments by other teams. Furthermore, Supplementary Figure 3 looks like a cut-paste composition of SDS gels line. Please indicate clear if you have not loaded all samples on the same gel and have cut lanes, e.g. lane of sample 5.

(10) Regarding the quantification of detergent (lines 535-537) and suppl. Fig. 5a, showing Bio-Bead removal of DM in a buffer solution but not from a liposomes. While this proves that Bio-Beads can absorb DM, this cannot be readily translated into a DM/liposome system. In a detergent/liposome system, the equilibration time of detergent through both leaflets of the liposome membrane play a major role in the detergent removal kinetics by Bio-Beads. In addition, note that the title of supplementary figure 5 might be misleading and I would recommend you add an extra sentence in suppl.fig.5a to clarify that the experiment is performed in buffer and thus, in the absence of liposomes.

Points to be clarified in the Materials and methods

(1) Related to point 8 of reviewer #3, please specify the temperature of the treatment with Bio-Beads and of the centrifugation (line 457). Same comment applies to lines 470-471, which is the temperature

used during Bio-Beads incubation? And in line 472, how long is the second round of incubation with Bio-Beads?

(2) In line 499, we would suggest changing the amount of protein from concentration to weight in order to avoid confusing whether this refers to the final concentration or to the starting amount added.

(3) Line 526-529, how many data points, for example in seconds, are considered to calculate FD, F0 and FT. Please describe.

Minor points:

(1) The authors use the abbreviation NBD for the protein domain and fluorescent lipids; these should be avoided.

(2) In figures where “x” and “o” appears, please change by “-” and “+”, respectively (e.g. in Figure 1a and 2e, among others).

(3) In panel and figure legend of Figure 3e, could the authors please indicate the location of the “lateral gate” mentioned in the main text (lines 277-278)?

(4) In the figure legend of figure 6f, I suggest to add “Lipid or detergent EM density” instead of only “EM density” to facilitate the reader to understand what it refers to.

(5) Supplementary Figure 2. Please indicate what the numbers on top of the SDS-Page in panel b indicate.

Comments from Reviewers (Bold) and Author Responses

Reviewer #1 (Remarks to the Author):

All points raised have been adequately addressed by the authors. I recommend acceptance of the revised version of the manuscript.

.....

Reviewer #2 (Remarks to the Author):

1. All the phospholipids used in the study should be carefully checked by means of two-dimensional TLC. If impurities and lyso-phospholipids are contaminated, side-effects may occur: lyso-phospholipids may affect the dispersion of micelle form, and act as detergent in membranous structure. Especially, transport assay could be prohibited by the detergent action of lyso-phospholipids. (Reviewer experienced that the labels had been stucked differently for vials of synthetic phospholipids from a famous lipid supplier, and so-called purified PS from another famous was found to be crude extract.)

>> As suggested, we have performed a TLC analysis to see whether the phospholipids used in this study were contaminated with lyso-phospholipids or other impurities. The 1D-TLC method was sufficient for us to observe distinct major spots of lyso-lipids which were well-separated from the natural lipids. We observed no contamination of our lipids, indicating that at least the TAPL-mediated lipid transport activity was not affected by the detergent-like properties of lyso-phospholipids. These data are shown in Supplementary Figure 6 and discussed on lines 181-184.

1) Authors did not answer the Comment 8 (2) as to NBD-lipids. Please describe the supplier of NBD-lipids or person donated, and their purity. How NBD is conjugated? It is described that NBD-lipids have a variety of acyl chain-labeled (Lines 147-148). However, it is not certain whether the labeling pattern is similar in different phospholipids or not. Please indicate these points in detail.

>> We have revised the text (lines 150-151) and Methods (lines 540-547), and added Supplementary Figure 4a, to clarify this point.

2) Although it may be true that the NBD-conjugation did not affect significantly on the ATPase activities of DM-purified Δ TMD0 (supplementary Fig.4b), it is too speculative if authors say that NBD-conjugation did not influence the functional properties of lipids as substrates of TAPL (Lines 148-149). Only floppase activity for NBD-PS was demonstrated in the revised manuscript.

>> We have revised the text (lines 150-151). As follows:

- before change

NBD conjugation did not influence the functional properties of the lipids as substrates of TAPL.

- after change

The conjugation of NBD* chromophore to one of the acyl chains of phospholipids did not alter the dependence of Δ TMD0 activity on phospholipids.

2. In contrast to original manuscript which showed that most lipids (PE, PS, PC, PG, sphingomyelin and cholesterol) flopped, only PS is demonstrated to move in the revised manuscript. The logic of identification of PG in the cryo-EM is exclusion of the possibility of PE and CL (Lines 248-250). However, it is also possible that phospholipid had been bound before solubilization of Δ TMD0.

>> We agree with this comment and have added the following sentence on lines 270-282.

“Because of the lipid promiscuity of Δ TMD0, it is possible that some endogenous lipids, such as PS or PC, co-purified with Δ TMD0 and occupied the upper cavity of the transporter without causing severe steric collision.”

Authors did not discuss the phospholipid and sterol contents in the lysosomal membrane. Biological meaning of phospholipid floppase activity of TAPL in lysosome should be discussed briefly although authors also showed the possibility that other phospholipids such as PS and PC could bind to lipid binding site (Lines 257-260). Useful information would be found in FEBS J. (2021) 288, 4168 and Int. J. Mol. Sci. (2019) 20, 2167.

>> The revised manuscript now includes a section discussing the biological implications of the lipid transport activity of TAPL in lysosomes. Please see lines 380-383.

3. Since the study is not only for the researchers of protein structure analyses but also many other fields such as biochemistry and medical sciences and so on, authors should mention their study carefully and politely (Comment 8) to those who are able to understand the story if this paper would become acceptable.

1) As for Fig.1, the Fig.1a is clear since transport assay is carried out using proteoliposome. The Fig1b is probably the results of DM-purified micelles from the legend (Line 644). Both DM micelles and nanodiscs are used in Fig1c as indicated in the legend (Line 646). However, how nanodiscs are prepared with PO (1-palmitoyl-2-oleoyl)-phospholipids (Fig.1c) were not indicated in the Materials and Methods (Lanes 427 – 443). The supplier and purity of PO-phospholipids were not mentioned. Please add these points.

>> We have revised the description in Methods, as suggested (lines 471-474).

2) As for Fig.2(a, b), detergent-purified (Line 132) and DM-purified (Lines 648, 655) Δ TMD0 are indicated. Since the title of Fig.2 is “transport activity” (Line 648), I would like to suggest that Fig.2(a, b) should be moved to Fig.1 (as d, e, respectively) or modify the figure title.

>> As suggested, Figures 2a and b have been renumbered as Figures 1d and e, respectively.

Is the “supplementary Fig.3” (Line 134) correct? Probably, supplementary Fig.4b is correct.

>> This was our mistake. Thank you for pointing this out.

>> In the revised version, this figure is cited on line 116.

Since supplementary Fig.4a is schematic of proteoliposome and title is flip-flop, appear of supplementary Fig.4b in the same figure is odd. The title of supplementary Fig.4 should be modified.

>> In the revised version, supplementary Fig. 4a becomes Fig. 2a.

>> As suggested, the title of supplementary Fig. 4 has been changed to “Attachment of the NBD* chromophore to the acyl chain of phospholipids has no significant effect on Δ TMD0 ATPase activity”.

Since supplementary Fig.3 (probably Fig.4b) is appeared together with Fig. 2a,b (Line134), the three small figures (Fig. 2a, b and supplementary Fig. 4b) may be the results of DM-purified Δ TMD0. However, this should be clearly indicated in the legend to supplementary Fig.4. Otherwise, the readers would misunderstand that the samples of supplementary Fig.4b were proteoliposomes like supplementary Fig. 4a. Supplementary Fig.3 should be cited in an appropriate position (probably Lines 107-121 or 427-443).

>> We have revised the legend to Supplementary Fig. 4b, as suggested.

>> Supplementary Fig. 3 is now cited in the main text. Please see line 116.

3) To Comment 6, not responded well: in above 1) and 2), DM-purified Δ TMD0 seems to be solubilized and purified without CHS. Please indicate whether the CHS is present or not during solubilization and purification in each sample used. It must be strengthened that the hysteresis strongly effects on the mode of interaction between membrane proteins and lipids.

>> We now provide this information throughout the article.

What kind of sample of Δ TMD0 (solubilized and purified in the presence CHS?) was

used to prepare nanodisc in the presence of CHS?

As for proteoliposome preparation, how was the purified protein (Lines 454-456, 469) prepared (solubilization and purification without CHS)?

>> We have clarified this point in Methods. See lines 475, 502-503 and 556.

As for cryo-EM sample preparation, authors indicated DM/CHS-purified Δ TMD0 (Line 567) and also Δ TMD0 purified in DM/CHS micelles (Line 201, 261). Probably Δ TMD0 was solubilized and purified in the presence of CHS. If so, authors should define Δ TMD0 as DM-purified Δ TMD0 (solubilized and purified in the absence of CHS) or DM/CHS-purified Δ TMD0 (or DM/CHS) (solubilized and purified in the presence of CHS) (Lines 416-422).

Meaning of the detergent-purified Δ TMD0 is not clear (Line 543). Does it mean DM-purified Δ TMD0 or DM/CHS-purified Δ TMD0?

>> To avoid confusion, we now refer to Δ TMD0 as DM- or DM/CHS-purified Δ TMD0 throughout the manuscript.

4) The Δ TMD0 in nanodisc without peptides and nucleotides was first subjected to cryo-EM analysis (Fig. 3 and Supplementary Fig. 6a-c). Then, authors analyzed DM/CHS micelles in the presence of 9-mer peptide. Further, they compared peptide bound structure with that without peptide in Supplementary Fig. 6d. In the legend to Supplementary Fig. 6d, it should be mentioned that cryo-EM structures of PG-bound nanodisc and CHS & peptide-bound DM/CHS micelles are compared.

>> We have added the following sentence to the legend of Supplementary Fig. 7d (now Supplementary Fig. 15), as suggested.

“The structures of PG-bound Δ TMD0 (in nanodisc) and CHS & peptide-bound Δ TMD0 (in DM/CHS micelle) are aligned.”

5) As for Supplementary Fig. 7a (Line 235), condition of samples for ATPase measurement is difficult to understand. Do authors use proteoliposomes, NBD-PS embedded-proteoliposomes, DM-micelles or nanodiscs? How do they add POPG and CHS?

>> We have revised the figure legend to clarify this point. We already describe how to prepare the lipid solution and add it to reaction mixture (Methods, lines 526-529).

4. Authors evaluated the leakiness of proteoliposomes as indicated in supplementary Fig. 5c. However, purpose and results of experiments of supplementary Fig. 5a-c were not explained. Please mention these essences in appropriate site (e.g. Lines 513-531).

>> We have re-written the text to discuss the results of Supplementary Figs. 5a-c. Please see lines 152-159.

Authors indicated the method of how to prepare NBD dye-labeled proteoliposomes in the original manuscript. However, in the revised manuscript this portion is omitted. Since authors made efforts to prepare the tightly sealed proteoliposomes for the assay using dithionite, why they include the important improved method in Method section. It is also necessary to add what kinds of NBD-lipid is added in the legend to supplementary Fig 5c. In Line 169, (Figs.2c, d) should be inserted.

>> As suggested, we have modified the Methods section to describe our efforts to generate non-leaky proteoliposomes. Please see page 24-25.

>> We changed the legend to Figs. 2c-d (now Figs. 2b-c) and supplementary Fig. 5c.

5. Lines 375-376: “the TAP complex has similar preferences to TAPL in terms of peptide length” is not correct. As the authors describe in Introduction (Lines 54 – 55), TAP complex favors peptides of 8-12 residues, whereas TAPL transports a broad spectrum of peptides, ranging from 6 to 59 residues. Although three reviewers did not indicate this part in the original manuscript, revision is necessary in the final version.

>> We agree that this sentence is confusing and have attempted to clarify it (Lines 409-411).

If difference in central region of peptide binding sites of TAP and TAPL could be imagined, please discuss.

>> Currently, only the cryo-EM structure of TAP bound to viral peptide ICP47 is available (Oldham et al. Nature 529, 2016, 537-540). As we believe that ICP47 binding may induce the deformation of the peptide binding site of the TAP complex, the proposed discussion is not currently possible. To address this question, one would need to determine the structure of the TAP complex in its apo state.

6. Lines 233-234: Briefly explain that the affinity of peptide binding to Δ TMD0 (wild and Y405) is determined by means of microscale thermophoresis.

>> We have done as suggested (line 253-255).

7. Lines 302-303: G452 of TAPL is also conserved in TAP2.

>> We have made this correction (line 325-326). Thank you for suggesting it.

8. As for peptide sequences (Comments 7 (2)), Please add also Ref.2 [JBC (2005) 280, 23631.] since peptide sequences [RRYQKSTEL and RRYQNSTC(Φ)L] were shown in the Ref.2 (Lines 98, 202).

>> We have added this reference.

=====

Reviewer #3 (Remarks to the Author):

The authors have resubmitted a substantially revised version of their previous manuscript with a number of new experiments added: lipid composition-dependent ATPase kinetics of Δ TMD0, lipid flip-flop activity of Δ TMD0 under non-ATP hydrolytic conditions (Na⁺/ATP), and flip-flop analysis of the ATPase-deficient E664Q mutant as a negative control. While the efforts of the authors are highly appreciated, there are still misleading statements and overinterpretations of the new results that should be addressed in a second revision as outline below.

(1) The authors measured now ATPase activity on both detergent-solubilized and nanodiscs-reconstituted TAPL. Essentially, all lipids tested stimulated the basal activity in both set-ups (PE could not be tested for technical reasons). Thus, the general lipid-dependent stimulation observed by the authors does not allow one to conclude on the substrate of TAPL. The author still over-interpret their observations and should carefully adjust the text and discussion of the manuscript. Here are only few examples of misleading statements throughout the revised manuscript:

a. Lines 74-76: The sentence reading “Moreover, to the best of our knowledge, this is the first report that TAPL binds directly to endogenous lipids and flips them from the cytoplasmic leaflet of the lysosome to the luminal leaflet in an ATP-dependent manner” states a function in living cells which has not been reported in the scope of this manuscript. Instead, we suggest replacing this sentence into something in the following lines: “Moreover, to the best of our knowledge, this is the first report that TAPL binds directly to endogenous lipids and transports a fluorescent lipid analogue in an ATP-dependent manner”.

>> We have made the change as suggested (Lines 74-77).

- as follows:

Moreover, to the best of our knowledge, this is the first report that TAPL exhibits lipid-dependent ATPase activity and transports fluorescently-labelled phosphatidylserine from the outer to the inner leaflet of the proteoliposome in an ATP-dependent manner.

b. The authors suggest a link between lipid stimulation and lipid transport, and state “The results suggest that lipids are TAPL substrates and their translocation is coupled to ATP hydrolysis (lines 140-141)”. The current data do not show this; all lipids stimulate ATPase activity but only NBD-PS is transported (without specific higher stimulation of ATPase activity). Along the lines, the title of Figure 2 is not precise: “Fig. 2. ATP-dependent lipid transport activity”; only in panel d, lipid transport is presented.

>> We have revised the text appropriately (page 7).

- before change

Together, these results support our hypothesis that lipids are TAPL substrates and their

translocation is coupled to ATP hydrolysis.

- after change

The stimulation of Δ TMD0 activity by lipid species supports our hypothesis that lipids are likely physiological substrates for TAPL. (Lines 142-143).

>> Figs. 2a and b have been moved to Figure 1 in response to Reviewer 1 (# point 3-2), so we have changed the title of Figure 2 to “ATP-dependent NBD*-PS transport activity”.

c. Statements such as “NBD conjugation does not impair the intrinsic lipid transport activity of Δ TMD0” are incorrect. The authors have only shown that NBD conjugation does not impair ATPase stimulation!

>> We have revised the text.

- before change

NBD conjugation does not impair the intrinsic lipid transport activity of Δ TMD0.

- after change

The conjugation of NBD* chromophore to one of the acyl chains of phospholipids does not impair the dependence of Δ TMD0 ATPase activity on phospholipids (Lines 149-151, Supplementary Fig. 4).

d. Lines 156-157: Based on the data from figure 2e, a more accurate interpretation would be to say that percentage of NBD-PS transport driven by a catalytically active Δ TMD0 was ~2%. This is based on that both in presence of the peptide or the catalytically inactive version also show a difference.

>> We have made the suggested change.

- before change

The net percentage of Δ TMD0-mediated NBD-PS translocation from the outer to the inner leaflet was ~4.5%.

- after change

The true active floppase activity of the transporter was ~2.6% (Lines 169-173).

(2) Line 151-151: “In the presence of Mg²⁺/ATP, the addition of 10 mM dithionite to Δ TMD-loaded proteoliposomes led to a fast drop in the fluorescence of NBD-PS to ~36% of the original value (Fig. 2c and Supplementary Figs. 5a, b)”

Supplementary Figs. 5a, b do not show this information given in the sentence. In fact, Supplementary Figs. 5a, b are not at all presented and discussed in the main text.

>> We have revised the text and Methods section to mention the results of Supplementary Figs. 5a and b. Please see pages 7-8 and 25.

(3) In supplementary Figure 5, panel c the authors use liposomes, not proteoliposomes. As such, the authors only probe the permeability of protein-free membranes; this should be clearly indicated in the figure, legend and text. In the current main text of the revised version, this panel is not presented and explain.

>> We have modified Supplementary Fig. 5 and the related text and legend as suggested.

(4) I cannot follow the argumentation of the authors on the different dithionite plateaus for liposomes and proteoliposomes. Why do not both vesicles have a 50/50% distribution of NBD-PS in the absence of ATP?

>> it has been reported that as in our dithionite-based experiments, in flip-flop assays using proteoliposomes reconstituted with hamster ABCB1 protein, NBD-lipids in protein-free liposomes lost 50 % of their fluorescence when treated with dithionite, whereas the insertion protein into the liposomes affected the distribution of the NBD-lipids between the leaflets of the membrane, such that more NBD-lipids were located in the outer leaflet (*Biochemistry* 2001, 40, 6937–6947). This finding explains why we detected a greater decrease of NBD fluorescence in proteoliposomes than in protein-free liposomes. We have re-written this section to clarify this point, and cite the reference mentioned above. Please see the lines 189-192.

Furthermore, Which NBD-lipid has been used, C6 or C12? Please give the full chemical name of the lipid probe.

>> We give the chemical structures and full names of all NBD-lipids used in this study in Supplementary Figure 4a and the Methods section (lines 540-547), respectively.

(5) Lipid transport activity was now tested along with crucial controls (ATPase-dead mutant, Na/Mg-ATP), showing that NBD-PS, a fluorescent analog of PS is transported. Whether TAPL transports endogenous lipids remains open. This should be clearly stated in the abstract and text of the manuscript. Strictly speaking, already the title overstates the data presented.

>> We have toned down the text (page 17) and title, as suggested.

(6) From supplementary Figure 4 it is unclear if the ATPase activity of Δ TMD0 have been measured on vesicle-reconstituted protein. Please indicate this clearly and consider revising the figure title.

>> We have modified the title and legend to Supplementary Figure 4 to clarify this point.

(7) Equal important, the authors should clear indicate the statistics to each panel in the main manuscript and in the supplement including the number of independent

experiments performed, i.e. independent reconstitutions. For example, the authors state that the data indicate that Δ TMD0-mediated lipid transport activity is significantly reduced under these conditions (Figures 2c-e.) but have not included the statistical analysis and significance, number of measurements, and number of independent reconstitutions.

>>We have revised the figure legends to clearly represent the statistics of the functional analysis.

(8) I could not find careful discussion on lipids/lipid translocation/floppase in relation of the structure and functional findings of Δ TMD0 in the Discussion section.

>> We now discuss the structure-function relationship of TAPL as suggested. Please see lines 380-383.

(9) Regarding the protein reconstitution, essential information is still missing. In Supplementary Figures 3 and 5, the authors should determine quantities of the protein based on the BSA standard. What was the final lipid to protein ratio that was achieved for the proteoliposomes. Such information is crucial for reproducing the experiments by other teams. Furthermore, Supplementary Figure 3 looks like a cut-paste composition of SDS gels line. Please indicate clear if you have not loaded all samples on the same gel and have cut lanes, e.g. lane of sample 5.

>> We have modified the legends to Supplementary Figs. 3 and 5 to indicate the final protein amounts associated with nanodiscs and liposomes, respectively.

>> We also revised the legend of Supplementary Figure 3 to indicate that the last lane was derived from a different SDS-PAGE gel.

(10) Regarding the quantification of detergent (lines 535-537) and suppl. Fig. 5a, showing Bio-Bead removal of DM in a buffer solution but not from a liposomes. While this proves that Bio-Beads can absorb DM, this cannot be readily translated into a DM/liposome system. In a detergent/liposome system, the equilibration time of detergent through both leaflets of the liposome membrane play a major role in the detergent removal kinetics by Bio-Beads. In addition, note that the title of supplementary figure 5 might be misleading and I would recommend you add an extra sentence in suppl.fig.5a to clarify that the experiment is performed in buffer and thus, in the absence of liposomes.

>> We now include quantitative data on the DM detergent adsorbed by Bio-Beads in both the buffer and liposome systems. This data is presented in Supplementary Fig. 5a and the modified legend.

Points to be clarified in the Materials and methods

(1) Related to point 8 of reviewer #3, please specify the temperature of the treatment with Bio-Beads and of the centrifugation (line 457). Same comment applies to lines 470-471, which is the temperature used during Bio-Beads incubation? And in line 472, how long is the second round of incubation with Bio-Beads?

>> We now provide this information in Methods lines 506, and 557-560.

(2) In line 499, we would suggest changing the amount of protein from concentration to weight in order to avoid confusing whether this refers to the final concentration or to the starting amount added.

>> We have made this change (Lines 520-524).

(3) Line 526-529, how many data points, for example in seconds, are considered to calculate FD, F0 and FT. Please describe.

>> We now give the numbers of data points used to calculate the average value for FD, F0 and FT (Line 577).

Minor points:

(1) The authors use the abbreviation NBD for the protein domain and fluorescent lipids; these should be avoided.

>> In the revised text, the fluorescent NBD chromophore attached to the lipids is referred to as NBD* to distinguish it from the nucleotide-binding domain (NBD) of the TAPL transporter.

(2) In figures where “x” and “o” appears, please change by “-“ and “+”, respectively (e.g. in Figure 1a and 2e, among others).

>> We have made these changes.

(3) In panel and figure legend of Figure 3e, could the authors please indicate the location of the “lateral gate” mentioned in the main text (lines 277-278)?

>> We have modified Figure 3d, (not 3e) to indicate the position of the lateral gate.

(4) In the figure legend of figure 6f, I suggest to add “Lipid or detergent EM density” instead of only “EM density” to facilitate the reader to understand what it refers to.

>> We have revised the legend to Figure 6f as suggested.

(5) Supplementary Figure 2. Please indicate what the numbers on top of the SDS-Page in panel b indicate.

>> We have done as suggested.

REVIEWERS' COMMENTS

Reviewer #2 (Remarks to the Author):

Authors well responded to all the comments. This time, I suggest acceptance of the re-revised version of the manuscript to Nature Communication. However, a few minor points indicated below should be still considered before printing.

1) Line 284: (Fig. 6d) -> (Fig. 4a and Fig. 6D)

Since sterol molecules are depicted in the Fig. 4a, please add "Fig. 4a".

2) Lines 786-787: please add (both from nanodisc-reconstituted Δ TMD0) to clear that the figure of inward-facing monomer is based on the data from nanodisc-reconstituted Δ TMD0.

3) Lines 421, 797: "a similar" is odd. "a similar to what?" Compared to structures of other ABC transporters?

Reviewer #3 (Remarks to the Author):

The authors have successfully integrated most of my points and more carefully interpreted their results. I have no further comments.

Comments from Reviewers (Bold) and Author Responses

Reviewer #2 (Remarks to the Author):

1) Line 284: (Fig. 6d) -> (Fig. 4a and Fig. 6D)

Since sterol molecules are depicted in the Fig. 4a, please add "Fig. 4a".

>> We have modified the text as suggested (line 284).

2) Lines 786-787: please add (both from nanodisc-reconstituted Δ TMD0) to clear that the figure of inward-facing monomer is based on the data from nanodisc-reconstituted Δ TMD0.

>> We have made the change as suggested (line 1038).

3) Lines 421, 797: "a similar" is odd. "a similar to what?" Compared to structures of other ABC transporters?

>> We have revised the text (line 419-421) and figure legend (line 797, now moved to line 1048) to clarify this point.

- before change

(line 421) In the resting state, the transporter may adopt a similar or more open inward-facing conformation that allows the substrates to easily approach their binding sites.

(line 797) In the resting state, TAPL probably has a similar or more open, inward-facing conformation for good access of substrates to their binding sites.

- after change

(line 419-421) In the resting state the transporter may adopt a conformation similar to that observed in lipid-bound TAPL or it may have a more open inward-facing conformation that allows the substrate to readily access the binding site.

(line 1048) In the resting state, TAPL probably has a conformation similar to that observed for lipid-bound TAPL or it may have a more open, inward-facing conformation for good access of substrates to their binding sites.